# Token Hidden Reward: Steering Exploration-Exploitation in Group Relative Deep Reinforcement Learning

**Wenlong Deng**[1,2], **Yi Ren**[1], **Yushu Li**[1], **Boying Gong**[4], **Danica J. Sutherland**[1,3],
**Xiaoxiao Li**[1,2†], **Christos Thrampoulidis**[1†]

[1]University of British Columbia, [2]Vector Institute, [3]Amii, [4]UC Berkeley
[†]Corresponding author

## Abstract

Reinforcement learning with verifiable rewards has significantly advanced the reasoning capabilities of large language models, yet how to explicitly steer training toward exploration or exploitation remains an open problem. We introduce Token Hidden Reward (THR), a token-level metric that quantifies each token's influence on the likelihood of correct responses under Group Relative Policy Optimization (GRPO). We find that training dynamics are dominated by a small subset of tokens with high absolute THR values. Most interestingly, tokens with positive THR strengthen confidence in correct outputs, thus favoring exploitation, while tokens with negative THR preserve probability mass for alternative outputs, enabling exploration. This insight suggests a natural intervention: a THR-guided reweighting algorithm that modulates GRPO's learning signals to explicitly bias training toward exploitation or exploration. We validate the efficacy of this algorithm on diverse math reasoning benchmarks. By amplifying tokens with positive THR value and weakening negative ones, our algorithm improves greedy-decoding accuracy, favoring exploitation. The reverse strategy yields consistent gains in Pass@K accuracy, favoring exploration. We further demonstrate that our algorithm integrates seamlessly with other RL objectives such as GSPO and generalizes across architectures including Llama. These findings establish THR as a principled and fine-grained mechanism for dynamically controlling exploration and exploitation in RL-tuned LLMs, providing new tools for targeted fine-tuning in reasoning-intensive applications.

## 1 Introduction

The integration of reinforcement learning with verifiable rewards (RLVR) has significantly advanced the reasoning capabilities of large language models (LLMs) (Guo et al., 2025; Jaech et al., 2024; Team et al., 2023). Group Relative Policy Optimization (GRPO) (Shao et al., 2024) and its variants (i.e., GSPO Zheng et al. (2025)) have emerged as a widely adopted and empirically successful method for training LLMs on complex reasoning tasks. Models like DeepSeek-R1 (Guo et al., 2025), DeepSeek-Math (Shao et al., 2024), Med-R1 (Lai et al., 2025), and Search-R1 (Jin et al., 2025) have leveraged GRPO to achieve state-of-the-art performance across diverse domains. Despite these successes, a central and persistent challenge in RL-driven LLM training is managing the inherent exploration-exploitation trade-off (Tang et al., 2024; Harris & Slivkins, 2025). Exploration, sampling uncertain actions to acquire novel information, is crucial for tasks demanding creativity (Lu et al., 2024) and enabling generalization to unseen test cases via scaling algorithms (Snell et al., 2024). Conversely, exploitation prioritizes optimal decision-making based on current knowledge, a preference in applications requiring high-confidence, low-variance responses, such as medical diagnosis (Wu et al., 2025). However, effectively shifting the training objective between exploration and exploitation remains an underexplored challenge.

Recent work has begun addressing this pressing challenge through various approaches. Chow et al. (2024) examine how to steer the balance between exploration and exploitation via a best-of-$n$ training objective, but their approach relies on an external verifier to select the best candidate

among $n$ generations. Contemporaneous works (Chen et al., 2025; Mahdavi et al., 2025; Walder & Karkhanis, 2025) introduce Pass@K-training to encourage exploration, though their methods primarily reweight questions based on hardness. Similarly, contemporaneous work (Cui et al., 2025) steers exploration by controlling entropy, but the analysis is limited to a token's influence on itself. In parallel, Deng et al. (2025) examines the learning dynamics of GRPO, showing how training alters the confidence of correct responses. By downweighting penalties on tokens that reduce this confidence, their method improves greedy decoding performance better exploiting model capabilities. However, their analysis is limited to negative gradients and their role in exploitation.

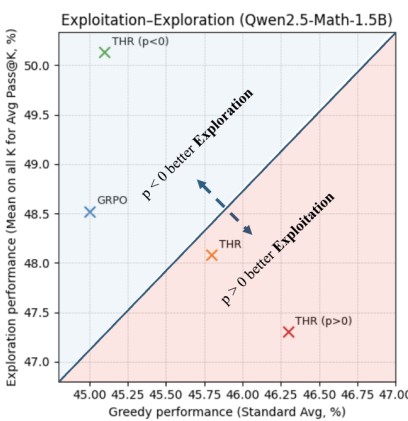

Motivated by Deng et al. (2025), we examine the intrinsic contribution of each token in the generated responses to the confidence of correct responses and connect this to the exploration–exploitation trade-off. We introduce Token Hidden Reward (THR), a token-level metric that quantifies how individual tokens influence the change in the likelihood of correct responses within the GRPO framework. Our analysis shows that a small subset of tokens carries disproportionately high absolute THR values, while most have negligible impact. Even more interestingly, leveraging the sign of THR, we design a reweighting strategy that explicitly adjusts learning signals : (1) **Positive THR tokens** amplify the likelihood change of correct responses, strengthening confidence and improving greedy decoding (*exploitation*); (2) **Negative THR tokens** preserves probability mass for alternative (than the correct) responses, boosting Pass@K performance (*exploration*). We specifically compare THR's token-level reweighting with question-level reweighting approaches such as Pass@K-training, showing that THR provides finer-grained and more effective guidance. Finally, we establish THR's theoretical and empirical connection to entropy-based exploration methods, while highlighting THR's efficiency in capturing cross-token interactions.

Figure 1: Our THR algorithm identifies high-influence tokens and reweights their learning signals based on sign: when $p > 0$, positive THR tokens are amplified (exploitation); when $p < 0$, negative THR tokens are amplified (exploration). The figure demonstrates control of exploration-exploitation trade-off.

In summary, our main contributions are threefold:

• We introduce Token Hidden Reward (THR) and conduct a thorough analysis, uncovering that a small subset of tokens disproportionately influences training and that the sign of THR correlates with the exploration-exploitation trade-off.

• We propose a THR-guided advantage reweighting strategy that effectively directs the fine-tuning process, enabling targeted emphasis on either exploitation or exploration. Fig. 1 for visualization.

• Empirical evaluations on math benchmarks confirm the effectiveness of THR-guided reweighting, resulting in the successful realization of desired performance improvements.

## 2 RELATED WORK

**Reinforcement Learning for LLM Reasoning.** Recent works have explored the use of model-generated solutions as a form of bootstrapping to strengthen the reasoning capabilities of large language models (LLMs)(Jaech et al., 2024; Guo et al., 2025; Team et al., 2025). These methods typically generate candidate solutions using a pre-trained model, then filter them based on intermediate correctness signals(Setlur et al., 2024) or final answer correctness (Guo et al., 2025; Team et al., 2025), producing high-quality data to train a new model. Building on the success of reinforcement learning from human feedback (RLHF) (Ouyang et al., 2022), follow-up works such as GRPO (Shao et al., 2024; Guo et al., 2025) use online training to further enhance reasoning. Moreover, reinforcement learning directly incorporates the model's incorrect outputs into training, which has been found to further boost reasoning performance (Seed et al., 2025). Despite these advances, the role of model-generated outputs during training remains underexplored.

**Optimizing for inference time objectives.** An increasing number of finetuning methods seek to align training with inference-time objectives. Some approaches treat inference-time computation as a flexible post-hoc design choice (Snell et al., 2024), while others explicitly optimize best-of-$n$ performance during training (Huang et al., 2025). The latter, however, depends on an external verifier

to select the best output, which complicates scalability. Another direction emphasizes exploitation: Deng et al. (2025) reduce penalties on tokens in incorrect responses that positively contribute to correct responses, thereby strengthening the model's most confident predictions. Their analysis, however, is restricted to negative gradients and does not address exploration. In parallel, several works focus on exploration. Pass@K training (Chen et al., 2025; Mahdavi et al., 2025; Walder & Karkhanis, 2025) encourages exploration by reweighting questions based on hardness, but operates only at the question level and overlooks token-level dynamics. Similarly, entropy-based regularization methods such as COV-KL (Cui et al., 2025) promote exploration by adjusting token entropy, yet they model only a token's self-influence. By contrast, our work directly targets token-level contributions and cross-token interactions, showing how they govern the exploration–exploitation balance in GRPO.

## 3 PRELIMINARY

**Notations.** $W$, $w_z$, and $h_z$ denote token unembedding matrix, unembedding of a token $z \in \mathcal{V}$, and hidden embedding of token-sequence $z \in \mathcal{V}^*$. $z_k$ is the $k$-th token in $z$ and $z_{<k}$ is the first $k-1$ tokens in $z$. For question $x$, the old policy $\pi_{\theta_{\text{old}}}$ generates a group of $G$ positive/negative responses $(\{y_i^+\}_{i \in [N^+]}, \{y_i^-\}_{i \in [N^-]})$ with $N^+ + N^- = G$. Lastly, $e_z \in \mathbb{R}^{|\mathcal{V}|}$ is one-hot vector for token $z$.

### 3.1 GROUP RELATIVE POLICY OPTIMIZATION

Group relative policy optimization, introduced in DeepSeek-Math (Guo et al., 2025) and DeepSeek-R1 (Shao et al., 2024), simplifies RLVR by eliminating the value function estimation required in PPO (Schulman et al., 2017). Instead of learning a separate value network, GRPO computes group-relative rewards within each training batch, reducing training complexity while maintaining stable policy updates. For a query pair $x$, the policy $\pi_\theta$ samples $G$ responses $\{y_i\}_{i=1}^G$. Each $y_i$ consists of a sequence of $|y_i|$ tokens. Given rewards $r_i \in \{0, 1\}$ for each response, GRPO computes normalized advantages $\hat{A}_{i,k} := \frac{r_i - \mu}{\sigma}$, where $\mu$ and $\sigma$ are the empirical average and standard deviation of the rewards. Specifically for binary rewards $r_i \in \{0, 1\}$, denoting $q = N^+/G$ the fraction of correct ($r_i = 1$) responses per group, GRPO's advantage scores become:

$$\hat{A}_{i,k} = \begin{cases} \sqrt{\frac{1-q}{q}} & \text{if } r_i = 1, \\ -\sqrt{\frac{q}{1-q}} & \text{if } r_i = 0. \end{cases} \tag{1}$$

Note that this is constant across all tokens $k = 1, \ldots, |y_i|$ in the $i$-th response. GRPO minimizes:

$$\mathbb{E}_{\substack{(x,a) \sim \mathcal{D} \\ \{y_i\}_{i=1}^G \sim \pi_{\theta_{\text{old}}}(\cdot | x)}} \left[ \frac{1}{\sum_{i=1}^G |y_i|} \sum_{i=1}^G \sum_{k=1}^{|y_i|} \min \left( \gamma_{i,k}(\theta) \hat{A}_{i,k}, \hat{A}_{i,k} \cdot \text{clip} \left( \gamma_{i,k}(\theta), 1 - \varepsilon, 1 + \varepsilon \right) \right) \right], \tag{2}$$

where $\varepsilon$ is a clipping hyperparameter, $\text{clip}(\cdot)$ is the clipping operation, and $\gamma_{i,k}(\theta) = \frac{\pi_\theta(y_{i,k}|x, y_{i,<k})}{\pi_{\theta_{\text{old}}}(y_{i,k}|x, y_{i,<k})}$ is the likelihood ratio between the current policy $\pi_\theta$ and the old policy $\pi_{\theta_{\text{old}}}$.

### 3.2 LIKELIHOOD CHANGE OF CORRECT RESPONSE IN GRPO

A recent study (Deng et al., 2025) analyzed the learning dynamics of GRPO, examining how the likelihood of correct responses $y_i^+$ evolves during training. They proved the following theorem using the unconstrained features framework (Yang et al., 2017; Mixon et al., 2022; Razin et al., 2024):

**Theorem 3.1.** *For any question $x$, at any time $t \geq 0$ of training, and any correct response $y_i^+, i \in [N^+]$, in addition to its dependence on the token unembeddings, the likelihood change $\frac{d}{dt} \ln \pi_{\theta(t)}(y_i^+|x)$ decreases as the following quantity increases:*

$$q^- \sum_{k=1}^{|y_i^+|} \sum_{j=1}^{N^-} \sum_{k'=1}^{|y_j^-|} \underbrace{\alpha_{k,k'}^- \cdot \langle \mathbf{h}_{x,y_{i,<k}^+}, \mathbf{h}_{x,y_{j,<k'}^-} \rangle}_{\text{Negative Token Hidden Reward}} - q^+ \sum_{k=1}^{|y_i^+|} \sum_{i'=1}^{N^+} \sum_{k''=1}^{|y_{i'}^+|} \underbrace{\alpha_{k,k''}^+ \cdot \langle \mathbf{h}_{x,y_{i,<k}^+}, \mathbf{h}_{x,y_{i',<k''}^+} \rangle}_{\text{Positive Token Hidden Reward}}. \tag{3}$$

*Here, the weights $\alpha_{k,k'}^\pm$ quantify the similarity of token-level prediction errors across responses:*

$$\alpha_{k,k''}^+ = \left\langle \mathbf{e}_{y_{i,k}^+} - \pi_{\theta(t)}(\cdot \mid x, y_{i,<k}^+), \mathbf{e}_{y_{i',k''}^+} - \pi_{\theta(t)}(\cdot \mid x, y_{i',<k''}^+) \right\rangle,$$

$$\alpha_{k,k'}^- = \left\langle \mathbf{e}_{y_{i,k}^+} - \pi_{\theta(t)}(\cdot \mid x, y_{i,<k}^+), \mathbf{e}_{y_{j,k'}^-} - \pi_{\theta(t)}(\cdot \mid x, y_{j,<k'}^-), \right\rangle.$$

where $q^+ = \sqrt{(1-q)/q}$, $q^- = \sqrt{q/(1-q)}$, and recall $q = N^+/G$.

This theorem provides the theoretical foundation of our analysis by explaining how individual tokens of both correct and incorrect responses influence training dynamics of correct response likelihood.

## 4 TOKEN HIDDEN REWARD

Using the log-likelihood change $\frac{d}{dt} \ln \pi_{\theta(t)}(\boldsymbol{y}_i^+ \mid \boldsymbol{x})$ as a proxy for the GRPO objective, we now introduce *Token Hidden Reward* (THR) to isolate and quantify each token's specific contribution to the model's confidence in correct outputs. We then establish how THR values encode exploration-exploitation dynamics in model training.

### 4.1 DEFINITION OF THR

**Definition 4.1.** Given a question $\boldsymbol{x}$ and a correct response $\boldsymbol{y}_i^+$, for any token $y_{j,k'}$, $k' \in [|\boldsymbol{y}_j|]$ in another (positive or negative) response $\boldsymbol{y}_j$, the THR quantifies that token's contribution to the change $\frac{d}{dt} \ln \pi_{\theta(t)}(\boldsymbol{y}_i^+ \mid \boldsymbol{x})$ in the likelihood of the correct response. Formally, the hidden reward for the $k'$-th token is defined as:

$$\mathrm{THR}(\boldsymbol{y}_i^+, \boldsymbol{y}_j, k') = (2r_j - 1) \cdot \sum_{k=1}^{|\boldsymbol{y}_i^+|} \alpha_{k,k'} \cdot \langle \mathbf{h}_{\mathbf{x}, \boldsymbol{y}_{i,<k}^+}, \mathbf{h}_{\mathbf{x}, \boldsymbol{y}_{j,<k'}} \rangle .$$

Note the negative sign for incorrect responses ($r(\boldsymbol{y}) = 0$) reflecting that GRPO penalizes those responses. In view of Theorem 3.1, a higher THR is associated with a larger increase in likelihood.

Since GRPO operates on groups of responses (thus, there can be multiple correct answers), we extend THR to the group setting by marginalizing over all positive responses:

**Corollary 4.2.** *Given a question $\boldsymbol{x}$ and the set of correct responses $\{\boldsymbol{y}_i^+\}_{N^+}$, for any token $y_{j,k'}$ in a response $\boldsymbol{y}_j$ (where $\boldsymbol{y}_j \in \{\boldsymbol{y}_i^+\}_{i \in [N^+]} \cup \{\boldsymbol{y}_i^-\}_{i \in [N^-]}$), the token hidden reward is defined as its contribution to the likelihood change of the group of correct responses $\sum_{i=1}^{N^+} \frac{1}{|\boldsymbol{y}_i^+|} \frac{d}{dt} \ln \pi_{\theta(t)}(\boldsymbol{y}_i^+ \mid \boldsymbol{x})$. Formally, the $k'$-th token's contribution to likelihood change of the group of correct responses is:*

$$\mathrm{THR}_{j,k'} \triangleq \mathrm{THR}(\boldsymbol{y}_j, k') \triangleq \sum_{i=1}^{N^+} \frac{1}{|\boldsymbol{y}_i^+|} \mathrm{THR}(\boldsymbol{y}_i^+, \boldsymbol{y}_j, k').$$

In Corollary 4.2, the magnitude of $\mathrm{THR}_{j,k'}$ quantifies the strength of each token's influence on the likelihood. The sign of $\mathrm{THR}_{j,k'}$ indicates whether a token positively or negatively contributes to the likelihood of generating the correct response.

### 4.2 CONNECTING THR WITH EXPLORATION AND EXPLOITATION.

Since the likelihood of correct responses reflects the model's confidence, we interpret changes in this likelihood, driven by token-level contributions, as signals of *exploitation* or *exploration*. In our context, we define these as follows:

*Exploration* is encouraged by a *lower* increase in the likelihood of observed correct responses since this preserves some probability mass for alternative outputs.

*Exploitation* is encourages by a *higher* increase in the likelihood of observed correct responses, since this strengthens confidence in those observed correct outputs.

Since THR values quantify the amount by which likelihood of correct responses increases, we can modulate the trade-off between exploration and

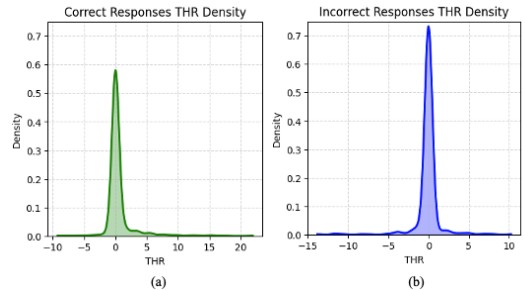

Figure 2: Density of THR scores for Qwen2.5-Math-1.5B. For both correct responses (a) and incorrect responses (b), we observe that only a small subset of tokens exhibits significantly high THR values. Notably, both types of responses contain tokens with both positive and negative THR scores.

exploitation through reweighting THR tokens: Amplifying positive THR tokens (by increasing their advantage weights) reduces the quantity in Eq. (3), boosting correct response likelihood and favoring exploitation. Conversely, amplifying negative THR tokens increases this quantity, reducing correct response likelihood and encouraging exploration. We validate these insights through our detailed analysis in Section 5 and exhaustive experiments in Section 6.

## 5 THR-GUIDED TOKEN ADVANTAGE ADJUSTMENT

In this section, we first analyze tokens' THR values and then propose a THR-based adjustment of token advantages to steer exploitation and exploration.

**THR Analysis.** Having defined THR, we now analyze its behavior in practice by examining the distribution of token-level THR scores in Fig. 2, where we observe:
_Dominant Tokens._ For both correct and incorrect responses, the majority of tokens have THR scores clustered around zero. However, a small subset of tokens exhibit significantly larger THR values, indicating that these tokens dominate the training dynamics.
_Sign of THR._ Both correct responses (a) and incorrect responses (b) contain tokens with both positive and negative THR scores, revealing that tokens in either response type can either strengthen or weaken confidence in correct outputs.
Then we use THR to guide the training from two complementary perspectives: **magnitude**, by focusing on the most influential tokens, and **sign**, by steering exploration versus exploitation.

**Dominant Token Training.** We define dominant tokens as those whose absolute THR score exceeds a threshold, i.e., $\text{THR} > \tau$. We detail the selection of $\tau$ in Section 6. To isolate the contribution of these tokens, we construct a training objective that masks out all others by setting their advantage to zero. The modified _token-level_ advantage becomes:

$$\hat{A}_{i,k}^{\text{THR}} = \mathbb{1}[|\text{THR}_{i,k}| > \tau] \cdot \hat{A}_{i,k}. \tag{4}$$

We refer to this setup as THR-only training. As shown in Table 1, this strategy achieves similar performance to the original GRPO method, which utilizes all tokens. This observation supports our claim that a small set of highly influential tokens largely determines performance.

**Steering Exploration and Exploitation via THR Sign.** To further exploit the information captured by THR, we introduce a token-level reweighting strategy that adapts training dynamics based on the sign of each token's THR score. Specifically, we modulate the advantage based on whether a token positively or negatively contributes to the correct response. To encourage exploitation, we increase the weight of tokens with positive THR and reduce that of tokens with negative THR. Conversely, to promote exploration, we reverse this weighting. This yields _token-level_ reweighted advantages:

$$\hat{A}_{i,k}^{\text{THR}(\text{p})} = \mathbb{1}[|\text{THR}_{i,k}| > \tau] \cdot (1 + \text{sign}(\text{THR}_{i,k}) \cdot p) \cdot \hat{A}_{i,k}. \tag{5}$$

When $p > 0$, this scheme boosts positive THR tokens while dampening negative THR tokens, thus reinforcing exploitation. In contrast, setting $p < 0$ reverses this behavior, shifting the training focus toward exploration. Experimental results for this reweighting approach are reported in Section 6.1. See also Fig. 1 for visualization of the tradeoff.

## 6 EXPERIMENTS & ANALYSIS

We evaluate THR's empirical effectiveness through comprehensive experiments across four dimensions: (1) Demonstrating exploitation ($p > 0$) and exploration ($p < 0$) capabilities as measured by greedy accuracy and Pass@K performance, (2) comparing our fine-grained token-level control against coarser-grained question-level baselines, (3) analyzing the relationship between THR and prediction entropy, and (4) validating generalizability across a GRPO variant (i.e., GSPO-token (Zheng et al., 2025) and Llama architectures.

**Experimental settings.** For all experiments, we follow Zeng et al. (2025) and train on the MATH dataset (levels 3–5) (Hendrycks et al., 2021). To accelerate training, we adopt _dynamic sampling_ (Yu et al., 2025), which discards samples with zero advantage and resamples until a full batch is formed. Unless otherwise specified, all models and methods are fine-tuned with identical reinforcement learning hyperparameters. Specifically, we use four A100 GPUs with a prompt batch size of 256 and 8 rollouts per prompt. We use a learning rate of $1\text{e}^{-6}$, and a mini-batch size of 64, resulting in 32 gradient updates per step. Training runs for 40 steps, which corresponds to more than two effective epochs given the higher throughput from dynamic sampling. We set the sampling temperature to

1.0, the clipping ratio to 0.2, and the KL loss coefficient to $1 \times 10^{-4}$. For the threshold $\tau$, we follow Deng et al. (2025, Eq. (8)), defining it as the average influence of the $i'$-th correct response's tokens on the likelihoods of other correct responses. Additional details are provided in Appendix B.

**Evaluation setup.** Since exploitation focuses on making the best decisions based on existing knowledge (Harris & Slivkins, 2025), we assess exploitation ability of fine-tuned models by measuring their greedy decoding accuracy. Here we adopt six widely used math benchmarks: three "*Hard datasets*" (AIME 2025, AIME 2024 (Veeraboina, 2023), AMC23) and three "*Standard datasets*" (MATH500 (Hendrycks et al., 2021), Olympiad (He et al., 2024), and Minerva Math (Lewkowycz et al., 2022)). To evaluate exploration, we report the unbiased Pass@$K$ accuracy (Chen et al., 2021) using temperature 1.0 on the challenging AIME2024, AIME2025 and AMC23 datasets, which require more exploration during attempts. The Pass@$K$ metric is defined as $\text{Pass}@K = \mathbb{E}_{x \sim D} \left[ 1 - \binom{M-C}{K} / \binom{M}{K} \right]$, where $M \geq K$ is the number of generated responses per question $x$, and $C$ denotes the number of correct responses. For all Pass@$K$ evaluations, we use $M = 256$ and report results for $K = 2^{1:8}$.

| Base Model | Method | Hard Datasets | | | | Standard Datasets | | | | Total Avg. |
|---|---|---|---|---|---|---|---|---|---|---|
| | | AIME25 | AIME24 | AMC23 | Hard Avg. | MATH500 | Minerva | Olympiad | Standard Avg. | |
| Qwen2.5-0.5B-Ins | Base | 0.0 | 0.0 | 2.5 | 0.8 | 33.4 | 4.4 | 7.0 | 14.9 | 7.9 |
| | GRPO | 0.0 | 0.0 | 7.5 | 2.5 | 33.8 | 8.8 | **9.9** | 17.5 | 10.0 |
| | THR | 0.0 | 0.0 | 15.0 | 5.0 | 34.6 | 8.1 | 7.6 | 16.8 | 10.9 |
| | THR ($p = -0.2$) | 0.0 | 0.0 | **20.0** | **6.7** | 34.0 | 9.9 | 8.9 | 17.6 | **12.1** |
| | THR ($p = 0.2$) | 0.0 | 0.0 | 17.5 | 5.8 | **35.6** | **11.0** | 6.5 | **17.7** | 11.8 |
| Qwen2.5-Math-1.5B | Base | 0.0 | 3.3 | 20.0 | 7.8 | 39.6 | 7.7 | 24.9 | 24.1 | 15.9 |
| | GRPO | 3.3 | **13.3** | 57.5 | 24.7 | **71.8** | 29.0 | 34.1 | 45.0 | 34.8 |
| | THR | 3.3 | **13.3** | 55.0 | 23.9 | 70.8 | 32.4 | 34.1 | 45.8 | 34.8 |
| | THR ($p = -0.1$) | **10.0** | **13.3** | 60.0 | **27.8** | 70.6 | 32.0 | 32.7 | 45.1 | **36.4** |
| | THR ($p = 0.1$) | 3.3 | **13.3** | **62.5** | 26.4 | 71.4 | **33.1** | **34.5** | **46.3** | 36.3 |
| Qwen2.5-Math-7B | Base | 13.3 | 6.7 | 42.5 | 20.8 | 64.6 | 15.8 | 26.7 | 35.7 | 28.3 |
| | GRPO | 13.3 | 10.0 | 62.5 | 28.6 | **82.2** | **46.0** | 42.1 | **56.8** | 42.7 |
| | THR | 10.0 | **16.7** | 65.0 | 30.6 | 80.8 | 44.1 | 43.1 | 56.0 | 43.3 |
| | THR ($p = -0.1$) | **23.3** | **16.7** | 62.5 | 33.9 | **82.2** | 36.8 | 42.4 | 53.8 | 44.0 |
| | THR ($p = 0.1$) | 20.0 | **16.7** | **75.0** | **37.2** | **82.2** | 43.4 | **43.4** | 56.3 | **46.8** |

Table 1: Exploitation Results on hard and standard math datasets. Pass@1 accuracy (%) using greedy decoding across different methods and datasets. **Bold** is best performance, underline is second-best.

| Method | Qwen2.5-0.5B-Instruct Pass@K | | | | | | | | | Qwen2.5-Math-1.5B Pass@K | | | | | | | | | Qwen2.5-Math-7B Pass@K | | | | | | | | |
|---|---|---|---|---|---|---|---|---|---|---|---|---|---|---|---|---|---|---|---|---|---|---|---|---|---|---|---|
| | 1 | 2 | 4 | 8 | 16 | 32 | 64 | 128 | 256 | 1 | 2 | 4 | 8 | 16 | 32 | 64 | 128 | 256 | 1 | 2 | 4 | 8 | 16 | 32 | 64 | 128 | 256 |
| **AIME 2025** | | | | | | | | | | | | | | | | | | | | | | | | | | | |
| Base | 0.1 | 0.2 | 0.3 | 0.6 | 1.2 | 2.5 | 5.0 | 10.0 | 20.0 | 1.3 | 2.6 | 4.9 | 8.6 | 13.9 | 19.9 | 26.2 | 33.4 | 40.0 | 2.7 | 5.0 | 8.9 | 14.7 | 21.7 | 29.5 | 37.4 | 44.5 | 50.0 |
| GRPO | **0.2** | **0.4** | **0.6** | 1.2 | **2.5** | **4.8** | **9.2** | 17.1 | 30.0 | 5.9 | 9.9 | 15.0 | 20.5 | 26.5 | 33.6 | 41.5 | 49.8 | 56.7 | 10.5 | 16.4 | 23.2 | 30.2 | 37.4 | 43.9 | 49.7 | 55.6 | 63.3 |
| THR | **0.2** | 0.3 | **0.6** | 1.2 | **2.5** | **4.8** | **9.2** | 17.1 | 30.0 | 5.4 | 9.2 | 14.1 | 19.4 | 25.0 | 31.7 | 39.5 | 48.0 | 56.7 | 9.6 | 15.2 | 21.9 | 29.2 | 36.2 | 42.6 | 49.8 | **58.3** | 63.3 |
| THR ($p < 0$) | **0.2** | 0.3 | **0.6** | 1.1 | 2.3 | 4.6 | 9.0 | **17.5** | **33.3** | **6.0** | **10.1** | **15.3** | **20.9** | 26.8 | **33.9** | 41.7 | 50.0 | **60.0** | **11.7** | **17.9** | **24.9** | **32.1** | **38.9** | **44.7** | **50.7** | 57.9 | **66.7** |
| THR ($p > 0$) | 0.1 | 0.3 | 0.5 | 0.9 | 1.9 | 3.7 | 7.3 | 14.2 | 26.7 | 4.6 | 8.0 | 12.8 | 18.7 | 25.6 | 33.7 | **43.0** | **52.5** | **60.0** | 9.3 | 15.2 | 22.4 | 29.9 | 36.5 | 42.4 | 48.5 | 55.9 | 63.3 |
| **AIME 2024** | | | | | | | | | | | | | | | | | | | | | | | | | | | |
| Base | 0.1 | 0.2 | 0.4 | 0.8 | 1.6 | 3.1 | 5.6 | 9.8 | 16.7 | 3.3 | 6.3 | 11.3 | 18.5 | 27.4 | 36.4 | 44.3 | 49.6 | 53.3 | 7.5 | 13.5 | 22.0 | 32.0 | 41.0 | 47.9 | 53.7 | 59.4 | 66.7 |
| GRPO | **0.4** | **0.8** | **1.5** | **2.9** | **5.4** | **10.0** | **17.2** | **27.3** | **36.7** | 11.4 | 17.7 | 24.3 | 30.5 | 36.7 | 43.4 | 50.0 | 56.0 | 63.3 | 14.4 | 20.7 | 27.5 | 34.7 | 42.0 | 49.6 | 58.1 | **67.3** | **76.7** |
| THR | **0.4** | 0.7 | **1.5** | **2.9** | **5.4** | 9.7 | 15.7 | 22.0 | 26.7 | 10.6 | 16.7 | 23.4 | 30.2 | 37.2 | 44.8 | 51.9 | 58.5 | 63.3 | 15.7 | 21.3 | 27.3 | 34.7 | **43.2** | **51.4** | 58.4 | 63.6 | 66.7 |
| THR ($p < 0$) | **0.4** | **0.8** | **1.5** | **2.9** | **5.4** | 9.4 | 14.9 | 21.5 | 30.0 | **11.9** | **18.2** | **24.9** | **31.2** | **37.9** | **45.3** | **52.9** | **61.2** | **70.0** | **17.3** | **22.6** | **28.5** | **35.5** | **43.2** | **51.3** | **58.8** | 66.3 | 73.3 |
| THR ($p > 0$) | **0.4** | 0.7 | 1.4 | 2.6 | 4.7 | 8.1 | 12.9 | 19.9 | 30.0 | 8.4 | 13.6 | 20.0 | 27.0 | 34.7 | 43.1 | 50.8 | 57.6 | 63.3 | 13.6 | 19.0 | 24.9 | 31.8 | 39.9 | 48.8 | 57.3 | 64.2 | 70.0 |
| **AMC23** | | | | | | | | | | | | | | | | | | | | | | | | | | | |
| Base | 4.1 | 7.8 | 14.0 | 23.4 | 36.1 | 50.6 | 64.4 | 75.4 | 82.5 | 15.3 | 26.7 | 42.1 | 58.6 | 72.3 | 81.9 | 88.8 | 94.3 | 97.5 | 25.0 | 40.6 | 58.2 | 72.9 | 82.8 | 88.7 | 92.6 | 96.2 | **100.0** |
| GRPO | 11.4 | 18.7 | 28.3 | 39.7 | 52.3 | 64.5 | 74.9 | 81.8 | 85.0 | 46.6 | 59.1 | 70.0 | 78.9 | 85.5 | 90.2 | 93.7 | 96.0 | 97.5 | **60.8** | **72.7** | **81.3** | 86.8 | 89.8 | 92.0 | 94.2 | 95.9 | 97.5 |
| THR | **12.0** | **20.2** | **30.8** | **43.0** | 56.1 | 68.6 | 79.5 | 88.0 | **92.5** | 44.8 | 57.8 | 69.1 | 78.2 | 85.1 | 90.1 | 93.6 | 95.9 | 97.5 | 58.1 | 71.3 | 80.7 | **87.1** | 90.9 | 93.5 | 95.9 | **98.3** | **100.0** |
| THR ($p < 0$) | **12.0** | 20.1 | 30.6 | 42.7 | **56.5** | **70.8** | **82.7** | **89.6** | **92.5** | **47.9** | **61.0** | **72.2** | **81.1** | **87.3** | **91.6** | **95.1** | **98.0** | **100.0** | 60.2 | 72.2 | 80.7 | 85.9 | 89.5 | 92.8 | 95.9 | **98.3** | **100.0** |
| THR ($p > 0$) | 11.1 | 18.8 | 29.2 | 41.9 | 56.0 | 69.3 | 80.1 | 87.5 | **92.5** | 41.4 | 54.8 | 66.8 | 76.6 | 84.2 | 89.5 | 93.2 | 95.8 | 97.5 | 57.0 | 70.0 | 79.8 | 86.8 | **91.2** | **94.0** | **96.1** | 97.3 | 97.5 |
| **Average** | | | | | | | | | | | | | | | | | | | | | | | | | | | |
| Base | 1.4 | 2.7 | 4.9 | 8.3 | 13.0 | 18.7 | 25.0 | 31.7 | 39.7 | 6.6 | 11.9 | 19.4 | 28.6 | 37.9 | 46.1 | 53.1 | 59.1 | 63.6 | 11.7 | 19.7 | 29.7 | 39.9 | 48.5 | 55.4 | 61.2 | 66.7 | 72.2 |
| GRPO | 4.0 | 6.6 | 10.1 | 14.6 | 20.1 | 26.4 | 33.8 | 42.1 | 50.6 | 21.3 | 28.9 | 36.4 | 43.3 | 49.6 | 55.7 | 61.7 | 67.3 | 72.5 | 28.6 | 36.6 | 44.0 | 50.6 | 56.4 | 61.8 | 67.3 | 72.9 | 79.2 |
| THR | **4.9** | **7.4** | **11.7** | **15.7** | 21.3 | 27.7 | 34.8 | 42.4 | 49.7 | 20.3 | 28.0 | 35.5 | 42.6 | 49.1 | 55.5 | 61.7 | 67.5 | 72.5 | 27.8 | 35.9 | 43.7 | 50.3 | 56.8 | 62.5 | 67.8 | 72.7 | 76.7 |
| THR ($p < 0$) | **4.9** | **7.4** | 11.6 | 15.6 | **21.4** | **28.3** | **35.5** | **43.5** | **51.9** | **21.9** | **29.8** | **37.5** | **44.4** | **50.7** | **57.3** | **63.2** | **69.7** | **76.7** | **29.7** | **37.6** | **44.7** | **51.2** | **57.2** | **62.9** | **68.5** | **74.2** | **80.0** |
| THR ($p > 0$) | **4.9** | 6.6 | 10.4 | 15.1 | 20.9 | 27.0 | 33.4 | 40.5 | 49.7 | 18.1 | 25.5 | 33.2 | 40.8 | 48.2 | 55.4 | 62.3 | 68.6 | 73.6 | 26.6 | 34.7 | 42.4 | 49.5 | 55.9 | 61.7 | 67.3 | 72.5 | 76.9 |

Table 2: Exploration Results. Pass@$K$ results for Qwen2.5-0.5B-Instruct, Qwen2.5-Math-1.5B, and Qwen2.5-Math-7B are reported on the AIME (24,25) and AMC23 datasets, along with their average.

## 6.1 EFFECTIVENESS OF THR IN EXPLOITATION AND EXPLORATION

We use varying-sized Qwen2.5 models (Yang et al., 2024): 0.5B-Ins, Math-1.5B, Math-7B.

**Impact of Dominant Tokens.** Training exclusively with THR-dominant tokens (Eq. (4)), results in performance comparable to original GRPO. In Table 1, vanilla THR ($p = 0$) matches GRPO in greedy accuracy across models. Similarly, in Table 2 it also performs on par with GRPO with respect to Pass@$K$. Thus, THR-dominant tokens play a critical role in guiding the training process.

**Exploitation ($p > 0$).** Setting $p > 0$ amplifies positive THR tokens while suppressing negative ones. As shown in Table 1, THR($p = 0.1$) increases the total average greedy accuracy over vanilla THR

($p = 0$) by 1.9% on Qwen2.5-Math-1.5B and 3.5% on Qwen2.5-Math-7B. It further outperforms GRPO by 1.1% and 4.0% on the same models, highlighting $p > 0$ as the most effective configuration for exploitation. Moreover, despite prioritizing exploitation, $p > 0$ maintains competitive Pass@K results at larger $K$, staying close to both vanilla THR and GRPO (Table 2).

**Exploration ($p < 0$).** To encourage exploration, we upweight tokens with negative THR values while down-weighting positive ones, leaving more probability mass for alternative generations. As shown in Table 2, $p < 0$ consistently delivers strong Pass@K performance across all model sizes. For example, on Qwen2.5-Math-1.5B, THR($p = -0.1$) surpasses the best baseline by 2.4% at Pass@128 and 5.0% at Pass@256, while Qwen2.5-Math-7B shows steady gains of about 1% on average across all $K$. In addition, $p < 0$ maintains competitive greedy accuracy, outperforming vanilla THR and GRPO on several datasets (Table 1). Although weaker than $p > 0$ on standard benchmarks, it excels on hard datasets such as AIME and AMC, with Qwen2.5-Math-1.5B even exceeding the $p > 0$ configuration. This suggests that allowing greater exploration can be beneficial for hard datasets.

## 6.2 THR vs. Pass@K Training: Token-Level vs. Question-Level Reweighting

**Pass@K Training as Question-Level Reweighting.** Chen et al. (2025); Mahdavi et al. (2025); Walder & Karkhanis (2025) develop RLVR objectives that directly target Pass@K optimization. For GRPO, these amount to re-weightings of the advantage scores in a way that favors "rare successes"—i.e., responses associated with "hard" questions. Crucially, the reweighting is uniform across all tokens and responses for a given question, which we term *question-level reweighting*. To be concrete, As we show in Appendix D.1, that Chen et al. (2025)'s question-level reweighting of vanilla GRPO advantages takes the following simplified form (assuming $G \geq K$):

$$\hat{A}_{i,k}^{@K} = \sqrt{\frac{\binom{N^-}{K}/\binom{G}{K}}{1 - \binom{N^-}{K}/\binom{G}{K}}} \cdot \sqrt{\frac{q}{1-q}} \cdot \hat{A}_{i,k}. \tag{6}$$

In practice, we adopt a convex combination $q \cdot \hat{A}_{i,k} + (1-q) \cdot \hat{A}_{i,k}^{@K}$ of vanilla GRPO advantage and the above Pass@K advantage, termed *Pass@K-mixed* (Chen et al., 2025), to avoid overly suppressing easy questions and preserve valuable learning signals. Empirically, Pass@K-mixed outperforms GRPO on both Qwen2.5-Math-1.5B (Table 3) and Llama3.2-3B-Instruct (Table 9). For training, we use $K = 4, G = 8$ throughout our experiments.

| Method | Qwen2.5-Math-1.5B Pass@K | | | | | | | | | Qwen2.5-Math-7B Pass@K | | | | | | | | |
|---|---|---|---|---|---|---|---|---|---|---|---|---|---|---|---|---|---|---|
| | 1 | 2 | 4 | 8 | 16 | 32 | 64 | 128 | 256 | 1 | 2 | 4 | 8 | 16 | 32 | 64 | 128 | 256 |
| **AIME 2025** | | | | | | | | | | | | | | | | | | |
| GRPO | 5.9 | 9.9 | 15.0 | 20.5 | 26.5 | 33.6 | 41.5 | 49.8 | 56.7 | 10.5 | 16.4 | 23.2 | 30.2 | 37.4 | 43.9 | 49.7 | 55.6 | 63.3 |
| Pass@K-mixed | 5.6 | 9.6 | 14.6 | 20.1 | 26.1 | 33.3 | **41.7** | 50.0 | 56.7 | 10.6 | 16.5 | 23.1 | 30.1 | 37.1 | 43.3 | 48.9 | 56.3 | **66.7** |
| THR ($p < 0$) | **6.0** | **10.1** | **15.3** | **20.9** | **26.8** | **33.9** | **41.7** | 50.0 | 60.0 | **11.7** | **17.9** | **24.9** | **32.1** | **38.9** | **44.7** | 50.7 | **57.9** | **66.7** |
| THR($p < 0$) +Passk-Mixed | 4.8 | 8.3 | 12.9 | 18.1 | 23.6 | 30.2 | 37.9 | 46.5 | 56.7 | 10.1 | 15.8 | 22.3 | 29.1 | 36.0 | 42.2 | 47.9 | 54.6 | 63.3 |
| THR($p < 0$)+χPassk+(1 − χ)GRPO | 5.7 | 9.6 | 14.4 | 19.3 | 24.7 | 31.9 | 40.9 | **51.2** | **63.3** | 11.1 | 17.4 | 24.7 | 31.9 | 38.4 | 44.6 | **50.9** | 57.2 | 63.3 |
| **AIME 2024** | | | | | | | | | | | | | | | | | | |
| GRPO | 11.4 | 17.7 | 24.3 | 30.5 | 36.7 | 43.4 | 50.0 | 56.0 | 63.3 | 14.4 | 20.7 | 27.5 | 34.7 | 42.0 | 49.6 | 58.1 | 67.3 | **76.7** |
| Pass@K-mixed | 10.6 | 16.7 | 23.5 | 30.3 | 37.1 | 44.3 | 51.2 | 57.5 | 63.3 | 14.9 | 20.7 | 26.8 | 33.8 | 41.2 | 49.1 | 58.0 | 67.9 | **76.7** |
| THR ($p < 0$) | **11.9** | **18.2** | **24.9** | **31.2** | **37.9** | **45.3** | **52.9** | **61.2** | **70.0** | 17.3 | 22.6 | 28.5 | 35.5 | 43.2 | 51.3 | 58.8 | 66.3 | 73.3 |
| THR($p < 0$) +Passk-Mixed | 10.4 | 16.5 | 23.4 | 30.0 | 36.4 | 41.8 | 49.8 | 59.0 | **70.0** | 13.7 | 19.4 | 25.7 | 33.2 | 41.6 | 49.8 | 57.3 | 64.8 | 73.3 |
| THR($p < 0$)+χPassk+(1 − χ)GRPO | 11.0 | 17.0 | 23.8 | 30.4 | 37.0 | 44.2 | 52.0 | 59.8 | 66.7 | **18.1** | **24.3** | **31.2** | **38.4** | **45.5** | **52.6** | **60.7** | **69.8** | **76.7** |
| **AMC23** | | | | | | | | | | | | | | | | | | |
| GRPO | 46.6 | 59.1 | 70.0 | 78.9 | 85.5 | 90.2 | 93.7 | 96.0 | 97.5 | 60.8 | 72.7 | **81.3** | **86.8** | 89.8 | 92.0 | 94.2 | 95.9 | 97.5 |
| Pass@K-mixed | 45.2 | 58.1 | 69.4 | 78.4 | 85.2 | 90.8 | 95.2 | 98.5 | **100.0** | 61.3 | **73.5** | **81.3** | 85.8 | 88.1 | 89.6 | 91.1 | 93.1 | 95.0 |
| THR ($p < 0$) | **47.9** | **61.0** | **72.2** | **81.1** | **87.3** | 91.6 | 95.1 | 98.0 | **100.0** | 60.2 | 72.2 | 80.7 | 85.9 | 89.5 | 92.8 | 95.9 | 98.3 | **100.0** |
| THR($p < 0$) +Passk-Mixed | 43.9 | 57.5 | 69.2 | 78.6 | 85.9 | 91.4 | 95.6 | 98.3 | **100.0** | 58.0 | 71.2 | 80.5 | 86.4 | **90.1** | **93.0** | **96.0** | **98.7** | **100.0** |
| THR($p < 0$)+χPassk+(1 − χ)GRPO | 46.8 | 59.6 | 70.6 | 79.4 | 86.4 | **91.8** | **95.8** | **98.6** | **100.0** | **61.4** | 72.3 | 80.2 | 85.3 | 88.8 | 92.0 | 95.1 | 97.1 | 97.5 |
| **Average** | | | | | | | | | | | | | | | | | | |
| GRPO | 21.3 | 28.9 | 36.4 | 43.3 | 49.6 | 55.7 | 61.7 | 67.3 | 72.5 | 28.6 | 36.6 | 44.0 | 50.6 | 56.4 | 61.8 | 67.3 | 72.9 | 79.2 |
| Pass@K-mixed | 20.5 | 28.1 | 35.8 | 42.9 | 49.5 | 56.1 | 62.7 | 68.7 | 73.3 | 28.9 | 36.9 | 43.7 | 49.9 | 55.5 | 60.7 | 66.0 | 72.4 | 79.5 |
| THR ($p < 0$) | **21.9** | **29.8** | **37.5** | **44.4** | **50.7** | **57.3** | **63.2** | 69.7 | **76.7** | 29.7 | 37.6 | 44.7 | 51.2 | 57.2 | 62.9 | 68.5 | 74.2 | **80.0** |
| THR($p < 0$)+Passk-Mixed | 19.7 | 27.4 | 35.2 | 42.2 | 48.6 | 54.5 | 61.1 | 67.9 | 75.6 | 27.3 | 35.5 | 42.8 | 49.6 | 55.9 | 61.7 | 67.1 | 72.7 | 78.9 |
| THR($p < 0$)+χPassk+(1 − χ)GRPO | 21.2 | 28.7 | 36.3 | 43.0 | 49.4 | 56.0 | 62.9 | **69.9** | **76.7** | **30.2** | **38.0** | **45.4** | **51.9** | **57.6** | **63.1** | **68.9** | **74.7** | 79.2 |

Table 3: Comparing exploration ability with Pass@$K$. Results for Qwen2.5-Math-1.5B and Qwen2.5-Math-7B are reported on the AIME 2024, AIME 2025, and AMC23 datasets, along with their average.

**THR as Token-Level Modification within a Question.** Contrasting to the question-level reweighting in Eq. (6), our THR algorithms in Eq. (4) and Eq. (5) operate at the *token-level* by reweighting the advantage with factors that are specific to tokens across responses within a question $x$. As formalized in Corollary 4.2, THR adjusts the advantage of each token based on whether it contributes positively or negatively to the likelihood. By setting $p < 0$ in Eq. (5), THR effectively reserves probability mass for alternative responses within the same question, thereby encouraging exploration.

**Comparing THR with Pass@K training.** We compare the performance of THR with $p < 0$ to Pass@K-mixed training. THR consistently outperforms Pass@K-mixed across all Pass@K metrics on Qwen models. With average improvement $> 1.1\%$ across most $K$ values on both Qwen2.5-Math-1.5B and Qwen2.5-Math-7B, this highlights THR's stronger ability to promote exploration.

**Directly combining THR with Pass@K training is Suboptimal.** We also investigate whether directly combining THR($p < 0$) with Pass@K-mixed yields additional benefits but found it underperforms compared to plain THR($p < 0$). We hypothesize that this is because Pass@K-mixed tends to assign excessively low weights to "easy" questions (for those, $N^-$ and thus the first reweighting factor in Eq. (6) is small), thereby weakening THR's ability to explore still-present and valuable token-level variations within them. To validate this hypothesis, we combine THR with a "static" version of Pass@K-mixed training where advantages become: $\chi \cdot \text{Pass@K} + (1 - \chi) \cdot \text{GRPO}$, for constant (question-independent) $\chi$. Setting $\chi = 0.2$ helps preserve the influence of easy questions. This modification leads to consistent improvements over THR($p < 0$)+Pass@K-mixed and even outperforms THR ($p < 0$) on Qwen2.5-Math-7B, with Pass@K performance increases by up to 0.7% for $K = 4, 8$ and shows steady gains across $K = 2^{1:7}$. These results suggest that while Pass@K training and THR target different aspects of exploration, maintaining adequate weight for easy questions allows THR to complement Pass@K training effectively.

In summary, both THR and Pass@K training employ what Chen et al. (2025) term implicit advantage design to steer exploration. However, THR provides more fine-grained control by operating at the token level, enabling more targeted and effective exploration management.

## 6.3 ON THE RELATION OF THR WITH ENTROPY

In this section, we study the relation between THR and entropy because entropy has long served as a proxy for exploration in RL (Wang et al., 2018; Cui et al., 2025).

| Method | Qwen2.5-Math-1.5B Pass@K | | | | | | | | | Qwen2.5-Math-7B Pass@K | | | | | | | | |
|---|---|---|---|---|---|---|---|---|---|---|---|---|---|---|---|---|---|---|
| | 1 | 2 | 4 | 8 | 16 | 32 | 64 | 128 | 256 | 1 | 2 | 4 | 8 | 16 | 32 | 64 | 128 | 256 |
| **AIME 2025** | | | | | | | | | | | | | | | | | | |
| Cov-KL | 5.3 | 9.1 | 14.0 | 19.4 | 25.1 | 31.4 | 37.8 | 44.2 | 50.0 | 11.5 | 17.5 | 24.1 | 30.8 | 37.6 | 43.6 | 48.9 | 54.2 | 60.0 |
| THR ($p < 0$) | 6.0 | 10.1 | 15.3 | 20.9 | 26.8 | 33.9 | 41.7 | 50.0 | 60.0 | 11.7 | 17.9 | 24.9 | 32.1 | 38.9 | 44.7 | 50.7 | 57.9 | 66.7 |
| **AIME 2024** | | | | | | | | | | | | | | | | | | |
| Cov-KL | 11.0 | 17.1 | 23.8 | 30.2 | 36.6 | 43.1 | 49.1 | 54.6 | 60.0 | 14.7 | 20.4 | 26.7 | 33.9 | 41.5 | 48.7 | 55.1 | 61.6 | 70.0 |
| THR ($p < 0$) | 11.9 | 18.2 | 24.9 | 31.2 | 37.9 | 45.3 | 52.9 | 61.2 | 70.0 | 17.3 | 22.6 | 28.5 | 35.5 | 43.2 | 51.3 | 58.8 | 66.3 | 73.3 |
| **AMC23** | | | | | | | | | | | | | | | | | | |
| Cov-KL | 46.8 | 59.3 | 70.3 | 79.3 | 86.1 | 91.2 | 94.8 | 96.8 | 97.5 | 62.3 | 73.5 | 81.4 | 86.7 | 89.9 | 92.2 | 94.5 | 96.2 | 97.5 |
| THR ($p < 0$) | 47.9 | 61.0 | 72.2 | 81.1 | 87.3 | 91.6 | 95.1 | 98.0 | 100.0 | 60.2 | 72.2 | 80.7 | 85.9 | 89.5 | 92.8 | 95.9 | 98.3 | 100.0 |
| **Average** | | | | | | | | | | | | | | | | | | |
| Cov-KL | 21.0 | 28.5 | 36.0 | 43.0 | 49.3 | 55.2 | 60.6 | 65.2 | 69.2 | 29.5 | 37.1 | 44.1 | 50.5 | 56.3 | 61.5 | 66.2 | 70.7 | 75.8 |
| THR ($p < 0$) | 21.9 | 29.8 | 37.5 | 44.4 | 50.7 | 57.3 | 63.2 | 69.7 | 76.7 | 29.7 | 37.6 | 44.7 | 51.2 | 57.2 | 62.9 | 68.5 | 74.2 | 80.0 |

Table 4: Comparing exploration ability with Pass@$K$. Results for Qwen2.5-Math-1.5B and Qwen2.5-Math-7B are reported on the AIME 2024, AIME 2025, and AMC23 datasets, along with their average.

**Dominant tokens overlaps with high entropy tokens.** For a confident (low-entropy) token $\mathbf{e}_{\mathbf{y}_{k'}} - \pi(\cdot | \mathbf{x}, \mathbf{y}_{<k'})$ has small magnitude, thus the resulting $\alpha_{\cdot, k'}$ in Definition 4.1 tends to be close to zero, leading to a low THR. We analyze the overlap between tokens with high THR scores and those with high entropy. For each sample, we select the same number of high-entropy tokens as high-THR tokens, compute their overlap rate, and plot the kernel density estimate (Chen, 2017) of the resulting overlap scores in Fig. 3. We find consistently high overlap ratio, often around 90%, indicating a strong correlation between THR and entropy. This finding is consistent with the observation of contemporaneous work (Wang et al., 2025), demonstrating that training on only the top 20% of high-entropy tokens is sufficient to achieve performance on par with GRPO using all tokens.

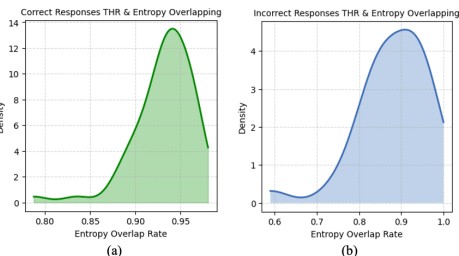

Figure 3: Overlap between high THR and high entropy tokens. For each sample, we quantify the overlap between tokens with high THR and high entropy, and plot the resulting density. The distribution shows a pronounced peak near 90%, highlighting a strong token-level association between these two metrics.

**Relation between THR and entropy regularization.** In Appendix D.2, we establish, under mild assumptions, a link between reweighting $p$ and entropy regularization at the token level. In particular, reweighting token advantages with THR implicitly regulates the dynamics of token entropy, with

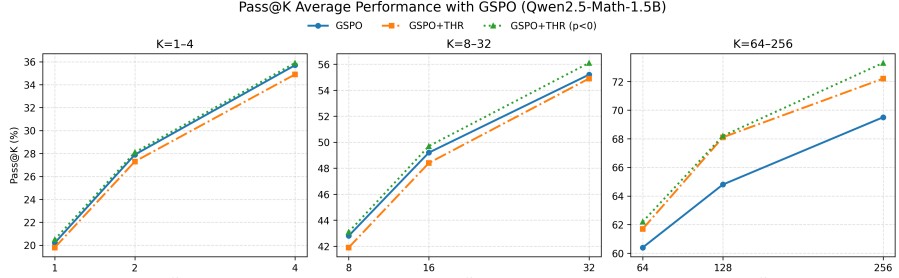

Figure 4: Mean of AIME 2024, AIME 2025, and AMC23 datasets' Pass@K performance of THR on GSPO using Qwen2.5-Math-1.5B across different K.

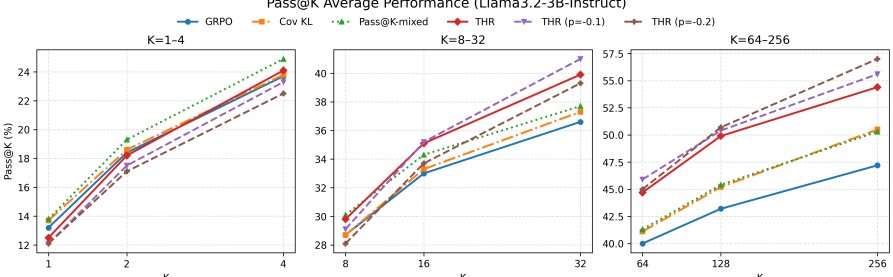

Figure 5: Mean of AIME 2024, AIME 2025, and AMC23 datasets' Pass@K performance of different methods on Llama3.2-3B-Instruct across different K.

both strength and direction determined by the hyper-parameter $p$[1]. Besides the conceptual similarity, we argue below that THR is a more efficient alternative to entropy-based methods.

**Comparison with COV-KL.** Cui et al. (2025) propose COV-KL as an entropy-based regularization approach focusing on how each token affects the update of itself during training. In contrast, THR, as formalized in Definition 4.1, explicitly captures the *cross-token* interactions that arise throughout the learning process. As shown in Table 4, THR($p < 0$) consistently outperforms COV-KL in all Pass@K settings, underscoring the importance of modeling cross-token influence for guiding exploration.

### 6.4 GENERALIZING THR TO OTHER RL OBJECTIVES AND MODEL FAMILIES

**Combining with other RL objectives.** We further show that THR can be seamlessly integrated with other group relative RL objectives. For demonstation, we apply THR to the token level variant of group sequence policy optimization (GSPO-token) (Zheng et al., 2025), which optimizes at the sequence level while allowing token level advantage adjustment (details in Appendix A). Fig. 4 shows that THR($p < 0$) boosts Pass@K performance across all K with an average improvement ∼0.9% to THR($p = 0$) and 1.4% to GSPO. See Apx. for detailed results.

**Performance on Llama.** To further demonstrate the generality of THR across model families, we evaluate it on Llama3.2-3B-Instruct. Unlike Qwen, Llama exhibits weaker mathematical knowledge, limited cognitive behaviors (Gandhi et al., 2025), and faces reduced reasoning length during training. Despite this, as shown in Fig. 5, THR still substantially boosts exploration, achieving up to a 7% Pass@K improvement compared to GRPO. Setting $p < 0$ amplifies these exploration gains even further. While baselines such as COV-KL and Pass@K-mixed also provide exploration improvements, they consistently underperform relative to THR. Reduced response length, results on exploitation, exploration results on each dataset, and more training details are provided in Appendix C.3.

## 7 CONCLUSION

We introduced THR, demonstrating that fine-grained analysis of learning dynamics can yield novel practical algorithmic insights steering exploration-exploitation in RLVR. Our findings suggest that RL for LLMs benefits from token-level interventions that leverage the unique structure of language generation, revealing new opportunities for principled algorithmic design. Our analysis connects THR with contemporaneous approaches, from Pass@K optimization's question-level reweighting to entropy-based exploration methods, reinforcing that multiple perspectives on the same underlying

---

[1]The strength and direction are controlled by the value and sign of hyper-parameter $p$

dynamics can complement and inform each other. As the field matures, combining insights from different analytical lenses (dynamics-based, entropy-based, objective-based) could yield even more sophisticated training methods. Specifically, our dynamics-first approach opens several promising directions itself, such as adaptive tuning of THR's parameter $p$ based on training progress or question difficulty and exploring similar token-level interventions in other RLVR domains from code generation to scientific reasoning.

**Acknowledgments:** This work was partially funded by the NSERC Discovery Grant RGPIN-2021-03677, Alliance Grant ALLRP 581098-22, the Natural Science and Engineering Research Council of Canada (NSERC), the Canada CIFAR AI Chairs program, the Canada Research Chair program, an IITP grant funded by MSIT, and the Digital Research Alliance of Canada. This work was partially funded under the Horizon Europe grant 101213369 DVPS.

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

CONTENTS

## A   ADDITIONAL PRELIMINARY

**Group Sequential Policy Optimization.** Recently, Zheng et al. (2025) introduce group sequence policy optimization (GSPO), a new reinforcement learning algorithm for training large language models. Following the basic principle of importance sampling, GSPO defines importance ratios based on sequence likelihood and performs sequence-level clipping, rewarding, and optimization. The GSPO objective $\mathcal{J}_{\text{GSPO}}(\theta)$ is then defined as:

$$\mathbb{E}_{\substack{(\boldsymbol{x},\boldsymbol{a})\sim\mathcal{D} \\ \{\boldsymbol{y}_i\}_{i=1}^{G}\sim\pi_{\theta_{\text{old}}}(\cdot|\boldsymbol{x})}} \left[ \frac{1}{\sum_{i=1}^{G}} \sum_{i=1}^{G} \min\left(s_i(\theta)\hat{A}_{i,k}, \hat{A}_{i,k}\cdot\text{clip}\left(s_i(\theta), 1-\varepsilon, 1+\varepsilon\right)\right) \right] \tag{7}$$

where the defined the importance ratio $s_i(\theta)$ is based on sequential likelihood:

$$s_i(\theta) = \left( \frac{\pi_\theta(\boldsymbol{y}_i|\boldsymbol{x})}{\pi_{\theta_{\text{old}}}(\boldsymbol{y}_i|\boldsymbol{x})} \right)^{\frac{1}{|\boldsymbol{y}_i|}} = \exp\left( \frac{1}{|\boldsymbol{y}_i|} \sum_{k=1}^{|\boldsymbol{y}_i|} \gamma_{i,k}(\theta) \right) \tag{8}$$

The token-level objective variant of GSPO, namely $\mathcal{J}_{\text{GSPO-token}}(\theta)$ allows token-wise advantage customization and is defined as:

$$\mathbb{E}_{\substack{(\boldsymbol{x},\boldsymbol{a})\sim\mathcal{D} \\ \{\boldsymbol{y}_i\}_{i=1}^{G}\sim\pi_{\theta_{\text{old}}}(\cdot|\boldsymbol{x})}} \left[ \frac{1}{G} \sum_{i=1}^{G} \frac{1}{|y_i|} \sum_{k=1}^{|y_i|} \min\left( s_{i,k}(\theta)\hat{A}_{i,k}, \text{clip}(s_{i,k}(\theta), 1-\epsilon, 1+\epsilon)\hat{A}_{i,k} \right) \right], \tag{9}$$

where

$$s_{i,k}(\theta) = \text{sg}[s_i(\theta)] \cdot \frac{\pi_\theta(\boldsymbol{y}_{i,k}|\boldsymbol{x},\boldsymbol{y}_{i,<k})}{\text{sg}[\pi_\theta(\boldsymbol{y}_{i,k}|\boldsymbol{x},\boldsymbol{y}_{i,<k})]}, \tag{10}$$

and sg[·] denotes only taking the numerical value but stopping the gradient, corresponding to the `detach` operation in PyTorch. The gradient of GSPO-token can be derived as:

GSPO demonstrates notably superior training stability, efficiency, and performance compared to GRPO and exhibits particular efficacy for the large-scale RL training of MoE models. To be specific,

## B   ADDITIONAL EXPERIMENT DETAILS.

**Additional Details for Qwen2.5-0.5B-Ins:** For the 0.5B model, training is conducted on two A6000 GPUs with a batch size of 32, a maximum rollout length of 2500 tokens, a learning rate of $5e^{-7}$, and a mini-batch size of 16—resulting in two iteration updates per training step. For the greedy decoding performance, we report the best accuracy across multiple checkpoints due to significant fluctuations during training. For all other settings, we report the performance at the final checkpoint. In addition to high-THR tokens, we also include those within the top 20% highest-entropy tokens that do not overlap with high-THR (approximate 4.1 % tokens), and keep their advantage unchanged being $\hat{A}_{i,k}$. For formatting, we follow Zeng et al. (2025), adopting simple prompts since the model struggles with complex instructions. We use $p = 0.2$ and $p = -0.2$ for exploitation and exploration respectively.

**Additional Details for Qwen-Math:** The Qwen-Math model Yang et al. (2024) uses its full context length of 3072 tokens for rollouts. For format, we follow Zeng et al. (2025) to use Qwen Chat template and require final answer to be enclosed in a latex command `\boxed{}`. Unless otherwise specified, we set $p = 0.1$ for exploitation and $p = -0.1$ for exploration.

| Base Model | Method | AIME25 | AIME24 | AMC23 | MATH500 | Minerva | Olympiad | Avg. |
|---|---|---|---|---|---|---|---|---|
| | Base | 0.0 | 3.3 | 20.0 | 39.6 | 7.7 | 24.9 | 15.9 |
| | GRPO | 3.3 | 13.3 | 57.5 | **71.8** | 29.0 | 34.1 | 34.8 |
| **Qwen2.5-Math-1.5B** | Pos Only | 3.3 | 10.0 | 57.5 | 70.6 | 30.1 | 31.0 | 33.8 |
| | THR ($p = 0.1$) | 3.3 | 13.3 | **62.5** | 71.4 | **33.1** | **34.5** | **36.3** |

Table 5: Exploitation Results. Pass@1 accuracy (%) using greedy decoding across different methods and datasets. **Bold** indicates the best performance, while underline marks the second-best.

**Additional Training Details for Llama:** For the Llama3.2-3B-Instruct Dubey et al. (2024) model, training is carried out on 8 A100 GPUs with a batch size of 256, a maximum rollout length of 3000 tokens, a learning rate of $1 \times 10^{-6}$, and a mini-batch size of 16. For greedy decoding, we report the best accuracy across multiple checkpoints due to the substantial fluctuations observed during training, while for all other settings we report results from the final checkpoint. In addition to high-THR tokens, we also include those within the top 20% highest-entropy tokens that do not overlap with high-THR (approximate 3.5 % tokens ), and fix their keep their advantage unchanged being $\hat{A}_{i,k}$. For formatting, we follow Zeng et al. (2025), adopting simple prompts since the model struggles with complex instructions.

## C  ADDITIONAL EXPERIMENTS

### C.1  ABLATION STUDY ON POSITIVE AND NEGATIVE-ONLY TRAINING.

We further investigate the impact of training with only positive or negative tokens by modifying $\hat{A}_{i,k}$. In the "Pos Only" setting, we set all values where $\hat{A}_{i,k} < 0$ to 0, thereby increasing the confidence of correct responses only. Conversely, in the "Neg Only" setting, we set all values where $\hat{A}_{i,k} > 0$ to 0, which reduces the confidence of incorrect responses without reinforcing correct ones. As shown in Table 5, "Pos Only" results in a 1.3% drop in average performance compared to GRPO, indicating that negative gradients also contribute to boosting confidence in correct responses.

| Method | Qwen2.5-0.5B-Instruct Pass@K | | | | | | | | | Qwen2.5-Math-1.5B Pass@K | | | | | | | | |
|---|---|---|---|---|---|---|---|---|---|---|---|---|---|---|---|---|---|---|
| | 1 | 2 | 4 | 8 | 16 | 32 | 64 | 128 | 256 | 1 | 2 | 4 | 8 | 16 | 32 | 64 | 128 | 256 |
| **AIME 2025** | | | | | | | | | | | | | | | | | | |
| GRPO | 0.2 | **0.4** | 0.6 | 1.2 | 2.5 | 4.8 | 9.2 | 17.1 | 30.0 | 5.9 | 9.9 | 15.0 | 20.5 | 26.5 | 33.6 | 41.5 | 49.8 | 56.7 |
| Neg Only | 0.2 | **0.4** | **0.7** | **1.4** | **2.8** | **5.3** | **9.5** | 16.2 | 26.7 | 4.7 | 8.1 | 12.7 | 17.8 | 23.4 | 30.2 | 38.2 | 46.2 | 56.7 |
| THR ($p < 0$) | **0.2** | 0.3 | 0.6 | 1.1 | 2.3 | 4.6 | 9.0 | **17.5** | **33.3** | **6.0** | **10.1** | **15.3** | **20.9** | **26.8** | **33.9** | **41.7** | **50.0** | **60.0** |
| **AIME 2024** | | | | | | | | | | | | | | | | | | |
| GRPO | **0.4** | **0.8** | **1.5** | **2.9** | **5.4** | **10.0** | **17.2** | **27.3** | **36.7** | 11.4 | 17.7 | 24.3 | 30.5 | 36.7 | 43.4 | 50.0 | 56.0 | 63.3 |
| Neg Only | 0.2 | 0.5 | 0.9 | 1.8 | 3.3 | 5.9 | 9.7 | 14.9 | 23.3 | 9.9 | 16.0 | 23.1 | 30.2 | 36.7 | 42.8 | 48.1 | 52.9 | 56.7 |
| THR ($p < 0$) | **0.4** | **0.8** | **1.5** | **2.9** | **5.4** | 9.4 | 14.9 | 21.5 | 30.0 | **11.9** | **18.2** | **24.9** | **31.2** | **37.9** | **45.3** | **52.9** | **61.2** | **70.0** |
| **AMC23** | | | | | | | | | | | | | | | | | | |
| GRPO | 11.4 | 18.7 | 28.3 | 39.7 | 52.3 | 64.5 | 74.9 | 81.8 | 85.0 | 46.6 | 59.1 | 70.0 | 78.9 | 85.5 | 90.2 | 93.7 | 96.0 | 97.5 |
| Neg Only | 7.7 | 13.7 | 22.6 | 34.4 | 48.4 | 63.2 | 76.6 | 87.5 | **95.0** | 44.0 | 56.9 | 68.0 | 76.5 | 83.0 | 88.5 | 92.3 | 94.3 | 95.0 |
| THR ($p < 0$) | **12.0** | **20.1** | **30.6** | **42.7** | **56.5** | **70.8** | **82.7** | **89.6** | 92.5 | **47.9** | **61.0** | **72.2** | **81.1** | **87.3** | **91.6** | **95.1** | **98.0** | **100.0** |
| **Average** | | | | | | | | | | | | | | | | | | |
| GRPO | 4.0 | 6.6 | 10.1 | 14.6 | 20.1 | 26.4 | 33.8 | 42.1 | 50.6 | 21.3 | 28.9 | 36.4 | 43.3 | 49.6 | 55.7 | 61.7 | 67.3 | 72.5 |
| Neg Only | 2.7 | 4.9 | 8.1 | 12.5 | 18.2 | 24.8 | 31.9 | 39.5 | 48.3 | 9.5 | 27.0 | 34.6 | 41.5 | 47.7 | 53.8 | 59.5 | 64.5 | 68.4 |
| THR ($p < 0$) | **4.9** | **7.4** | **11.6** | **15.6** | **21.4** | **28.3** | **35.5** | **43.5** | **51.9** | **21.9** | **29.8** | **37.5** | **44.4** | **50.7** | **57.3** | **63.2** | **69.7** | **76.7** |

Table 6: Comparing exploration ability with Pass@$K$. Results for Qwen2.5-Math-1.5B and Qwen2.5-Math-7B are reported on the AIME 2024, AIME 2025, and AMC23 datasets, along with their average. **Bold** indicates the best performance.

As also shown in Table 6, "Neg Only" underperforms in most cases. For example, on AMC23 with Qwen2.5-Math-1.5B, it achieves a Pass@256 of 56.7%, compared to 63.3% for both GRPO and vanilla THR. While "Neg Only" yields moderate improvements over the Base model on average—indicating that suppressing incorrect responses provides some exploratory value—positive tokens still play a critical role in enhancing exploration. By selectively incorporating informative tokens, THR with $p < 0$ achieves substantially better exploration performance than "Neg Only" alone.

| Method | Qwen2.5-Math-1.5B Pass@K | | | | | | | | |
|---|---|---|---|---|---|---|---|---|---|
| | 1 | 2 | 4 | 8 | 16 | 32 | 64 | 128 | 256 |
| **AIME 2025** | | | | | | | | | |
| GSPO | 5.2 | 9.0 | 13.9 | 19.3 | 24.9 | 31.0 | 36.9 | 41.4 | 46.7 |
| GSPO+THR | 4.4 | 7.8 | 12.5 | 18.0 | 23.9 | 31.1 | 39.0 | 46.4 | 50.0 |
| GSPO+THR ($p = -0.1$) | 5.1 | 8.9 | 14.3 | 20.4 | 26.6 | 33.3 | 39.9 | 46.9 | 53.3 |
| **AIME 2024** | | | | | | | | | |
| GSPO | 10.4 | 16.8 | 24.1 | 31.3 | 38.5 | 45.6 | 52.4 | 59.4 | 66.7 |
| GSPO+THR | 10.0 | 16.2 | 23.6 | 30.8 | 37.7 | 44.8 | 52.8 | 60.8 | 66.7 |
| GSPO+THR ($p = -0.1$) | 11.0 | 17.2 | 24.2 | 31.0 | 37.8 | 44.9 | 51.8 | 59.1 | 66.7 |
| **AMC 2023** | | | | | | | | | |
| GSPO | 44.9 | 58.0 | 69.0 | 77.7 | 84.3 | 89.1 | 92.0 | 93.6 | 95.0 |
| GSPO+THR | 44.9 | 58.0 | 68.7 | 77.0 | 83.5 | 88.8 | 93.3 | 97.2 | 100.0 |
| GSPO+THR ($p = -0.1$) | 45.4 | 58.2 | 69.1 | 77.9 | 84.6 | 90.1 | 95.0 | 98.7 | 100.0 |
| **Average** | | | | | | | | | |
| GSPO | 20.2 | 27.9 | 35.7 | 42.8 | 49.2 | 55.2 | 60.4 | 64.8 | 69.5 |
| GSPO+THR | 19.8 | 27.3 | 34.9 | 41.9 | 48.4 | 54.9 | 61.7 | 68.1 | 72.2 |
| GSPO+THR ($p = -0.1$) | **20.5** | **28.1** | **35.9** | **43.1** | **49.7** | **56.1** | **62.2** | **68.2** | **73.3** |

Table 7: Performance with GSPO

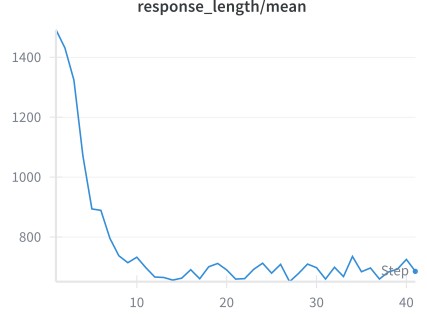

Figure 6: Response length dynamics of Llama3.2-3B-Instruct across different stages of GRPO training.

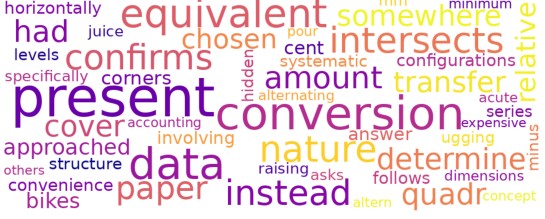

Figure 7: Word cloud of the top 50 tokens ranked by THR, generated from Qwen2.5-Math-7B on AMC23. Font size is proportional to each token's average THR. Tokens with high THR represent the key reasoning steps most critical in the model's problem-solving process.

## C.2 ADDITIONAL RESULTS ON GSPO

We further show that THR can be seamlessly integrated with other group relative reinforcement learning objectives. In particular, we apply THR to token level variant of group sequence policy optimization (GSPO-token) Zheng et al. (2025), which optimizes at the sequence level through clipping, rewarding, and optimization while allow token level advantage adjustment (more details in Appendix Appendix A). As reported in Table 7, incorporating THR with $p < 0$ yields substantial improvements, boosting Pass@K performance across all K with an average improvement by around 0.9% to THR and 1.4% to GSPO.

## C.3 ADDITIONAL RESULTS ON LLAMA.

**Reduced response length.** As shown in Fig. 6, the response length of Llama3.2-3B declines rapidly after a few epochs, with the average length dropping from about 1.5K tokens to roughly 650. This reduction may stem from the model's limited cognitive behaviors Gandhi et al. (2025). **Exploitation Results on Llama** We report the greedy decoding performance of Llama in Table 8. As shown in table, while GRPO achieves the best performance, setting $p > 0$ can improve the greedy decoding performance compared with vanilla THR by 1.1%.

| Base Model | Method | AIME25 | AIME24 | AMC23 | MATH500 | Minerva | Olympiad | Avg. |
|---|---|---|---|---|---|---|---|---|
| | Base | 0.0 | 3.3 | 22.5 | 40.2 | 16.5 | 11.9 | **15.7** |
| | GRPO | 0.0 | 26.7 | 30.0 | 54.4 | 22.1 | 18.1 | **25.2** |
| Llama3.2-3B-Instruct | THR | 0.0 | 13.3 | 32.5 | 51.8 | 22.1 | 19.9 | 23.3 |
| | THR ($p = -0.2$) | 3.3 | 6.7 | 27.5 | 51.4 | 20.6 | 16.3 | 21.0 |
| | THR ($p = 0.05$) | 3.3 | 13.3 | 40.0 | 50.6 | 22.4 | 16.7 | 24.4 |

Table 8: Exploitation Results. Pass@1 accuracy (%) using greedy decoding across different methods and datasets. **Bold** indicates the best performance, while underline marks the second-best.

| Method | Llama3.2-3B-Instruct Pass@K | | | | | | | | |
|---|---|---|---|---|---|---|---|---|---|
| | 1 | 2 | 4 | 8 | 16 | 32 | 64 | 128 | 256 |
| **AIME 2025** | | | | | | | | | |
| Base | 0.2 | 0.3 | 0.6 | 1.2 | 2.4 | 4.6 | 8.45 | 14.2 | 20.0 |
| GRPO | 0.3 | 0.7 | 1.25 | 2.4 | 4.3 | 7.0 | 10.2 | 13.2 | 16.7 |
| Cov KL | 0.4 | 0.7 | 1.4 | 2.5 | 4.5 | 7.4 | 11.2 | 16.3 | 23.3 |
| Pass@K-mixed | 0.7 | 1.3 | 2.3 | 3.9 | 6.3 | 9.1 | 12.6 | 16.7 | 20.0 |
| THR | 1.0 | 1.8 | 3.4 | 5.7 | 8.6 | 12.0 | 16.7 | 24.0 | 30.0 |
| THR ($p = -0.1$) | 1.1 | 2.1 | 3.8 | 6.7 | 10.7 | 15.3 | 19.7 | 24.2 | 30.0 |
| THR ($p = -0.2$) | 0.5 | 0.9 | 1.8 | 3.4 | 6.4 | 11.1 | 17.8 | 26.3 | 36.7 |
| **AIME 2024** | | | | | | | | | |
| Base | 1.4 | 2.6 | 4.8 | 8.3 | 13.4 | 20.3 | 28.4 | 35.9 | 40.0 |
| GRPO | 12.7 | 17.5 | 22.4 | 27.4 | 31.0 | 33.3 | 34.9 | 36.7 | 40.0 |
| Cov KL | 11.9 | 15.9 | 20.4 | 25.6 | 30.6 | 33.8 | 35.8 | 38.3 | 43.3 |
| Pass@K-mixed | 12.2 | 17.2 | 22.4 | 27.4 | 30.8 | 32.8 | 35.1 | 38.2 | 43.3 |
| THR | 9.8 | 15.0 | 20.5 | 25.7 | 29.8 | 32.6 | 35.0 | 38.2 | 43.3 |
| THR ($p = -0.1$) | 9.2 | 13.9 | 19.0 | 24.2 | 29.3 | 33.5 | 36.5 | 40.0 | 46.7 |
| THR ($p = -0.2$) | 9.4 | 13.6 | 18.2 | 23.1 | 27.9 | 32.5 | 37.1 | 41.6 | 46.7 |
| **AMC 2023** | | | | | | | | | |
| Base | 9.6 | 17.0 | 27.7 | 41.0 | 55.7 | 69.2 | 80.1 | 86.4 | 90.0 |
| GRPO | 26.7 | 36.9 | 47.3 | 56.4 | 63.6 | 69.5 | 74.8 | 79.6 | 85.0 |
| Cov KL | 28.9 | 39.3 | 49.6 | 57.9 | 64.7 | 70.8 | 76.2 | 81.1 | 85.0 |
| Pass@K-mixed | 28.6 | 39.3 | 49.9 | 58.9 | 65.8 | 71.3 | 76.3 | 81.4 | 87.5 |
| THR | 26.8 | 37.9 | 48.5 | 57.9 | 67.0 | 75.2 | 82.3 | 87.5 | 90.0 |
| THR ($p = -0.1$) | 26.1 | 36.4 | 47.0 | 56.4 | 65.5 | 74.2 | 81.5 | 87.0 | 90.0 |
| THR ($p = -0.2$) | 26.5 | 36.7 | 47.6 | 57.8 | 66.9 | 74.4 | 80.2 | 84.3 | 87.5 |
| **Average** | | | | | | | | | |
| Base | 3.7 | 6.6 | 11.0 | 16.8 | 23.8 | 31.4 | 39.0 | 45.5 | 50.0 |
| GRPO | 13.2 | 18.4 | 23.7 | 28.7 | 33.0 | 36.6 | 40.0 | 43.2 | 47.2 |
| Cov KL | 13.7 | 18.6 | 23.8 | 28.7 | 33.3 | 37.3 | 41.1 | 45.2 | 50.5 |
| Pass@K-mixed | 13.8 | 19.3 | 24.9 | 30.1 | 34.3 | 37.7 | 41.3 | 45.4 | 50.3 |
| THR | 12.5 | 18.2 | 24.1 | 29.8 | 35.1 | 39.9 | 44.7 | 49.9 | 54.4 |
| THR ($p = -0.1$) | 12.1 | 17.5 | 23.3 | **29.1** | **35.2** | **41.0** | **45.9** | 50.4 | 55.6 |
| THR ($p = -0.2$) | 12.1 | 17.1 | 22.5 | 28.1 | 33.7 | 39.3 | 45.0 | **50.7** | **57.0** |

Table 9: Pass@K performance of different methods using Llama3.2-3B-Instruct .

**Exploration Results on Llama** As shown in Table 9, THR still substantially boosts exploration, achieving over a 7% Pass@K improvement compared to GRPO. Setting $p < 0$ amplifies these exploration gains even further. While baselines such as COV-KL and Pass@K-mixed also provide exploration improvements, they consistently underperform relative to THR.

## C.4 ADDITIONAL THR TOKEN ANALYSIS

We further analyze tokens with high THR values using a word cloud visualization, as shown in Figure 7. The representative tokens can be organized into five functional categories that correspond to step-by-step reasoning:

**Stating the Given Information**: tokens that capture the initial conditions or input facts (*present, data, paper*).

**Transformation and Operations**: tokens that describe conversions, equivalence, or transfers of knowledge (*conversion, transfer, equivalent*).

**Constraints and Relationships**: tokens indicating dependencies, limitations, or structural relations (*relative, intersects, amount, dimensions*).

**Decision and Selection**: tokens reflecting choices among alternatives or branching reasoning paths (*determine, instead, alternating, altern, others*).

**Verification and Conclusion**: tokens signaling validation or consolidation of results (*confirms, systematic, answer*).

## C.5 RUNNING TIME OF EACH MODULE.

We also track the average time cost of each module during training, as reported in Table 10. Notably, the data generation (Data Gen) module that using dynamic sampling accounts for the majority of the total training time. In contrast, the overhead introduced by THR is minimal, e.g. 37 seconds for Qwen2.5-Math-1.5B, contributing only a small fraction to the overall cost.

| Model+dataset | Data Gen | Model Upd | THR | Ref | Old Prob | Total (Sec) |
|---|---|---|---|---|---|---|
| Qwen2.5-Math-1.5B | 347 | 210 | 37 | 120 | 120 | 834 |
| Qwen2.5-Math-7B | 422 | 371 | 39 | 187 | 187 | 1206 |
| Llama3.2-3B-Instruction | 625 | 139 | 26 | 89 | 89 | 968 |

Table 10: Average running time (per step, in seconds) of each module for different models and tasks.

## D DETAILED PROOFS

### D.1 PASS@K AS THE QUESTION LEVEL REWEIGHTING

Chen et al. (2025); Mahdavi et al. (2025); Walder & Karkhanis (2025) develop RLVR objectives that directly target Pass@K optimization. Starting with GRPO's ancestor, REINFORCE, Mahdavi et al. (2025); Walder & Karkhanis (2025) derive reward rescalings by directly optimizing the Pass@K objective. Mahdavi et al. (2025) apply the same rescaling to advantages giving a GRPO version of their approach. These rescalings upweight the gradient contribution of correct responses that constitute "rare successes"—i.e., responses associated with "hard" questions. Crucially, the reweighting is uniform across all tokens and responses for a given question, which we term *question-level reweighting*. More recently, Chen et al. (2025) introduce an appealing alternative to optimizing Pass@K by incorporating the design directly within GRPO's group structure. Here, we simplify the formulas in Chen et al. (2025) and arrive at an explicit formulation of advantage shaping that reveals its question-level nature. Starting from the defined advantages in Chen et al. (2025):

$$\bar{R}^{\text{group}} = 1 - \frac{\binom{N^-}{K}}{\binom{G}{K}}, \sigma^{\text{group}} = \sqrt{\bar{R}^{\text{group}} \times (1 - \bar{R}^{\text{group}})}$$

$$A_{\text{pos}}^{@K} = \frac{1 - \bar{R}^{\text{group}}}{\sigma^{\text{group}}}, A_{\text{neg}}^{@K} = \left(1 - \bar{R}^{\text{group}} - \frac{\binom{N^- - 1}{K - 1}}{\binom{G - 1}{K - 1}}\right) \times (\sigma^{\text{group}})^{-1}.$$

Since $N^- = (1 - q)G$ then we can obtain:

$$A_{\text{pos}}^{@K} = \frac{\binom{N^-}{K}}{\binom{G}{K}\sigma^{\text{group}}}$$

$$= \frac{\prod_{i=0}^{K-1}((1-q)G - i)}{\prod_{i=0}^{k-1}(G - i)\sigma^{\text{group}}},$$

$$= \sqrt{\frac{\binom{N^-}{K}/\binom{G}{K}}{1 - \binom{N^-}{K}/\binom{G}{K}}}$$

$$= \sqrt{\frac{\binom{N^-}{K}/\binom{G}{K}}{1 - \binom{N^-}{K}/\binom{G}{K}} \cdot \sqrt{\frac{q}{1-q}}} \cdot \sqrt{\frac{1-q}{q}}$$

$$= \sqrt{\frac{\binom{N^-}{K}/\binom{G}{K}}{1 - \binom{N^-}{K}/\binom{G}{K}} \cdot \sqrt{\frac{q}{1-q}}} \cdot \hat{A}_{\text{pos}} \tag{11}$$

then harder question will have a larger $1 - q$ thus larger advantage, then we derive the negative advantage.

$$
\begin{aligned}
A_{\text{neg}}^{@K} &= \left(\frac{\binom{N^-}{K}}{\binom{G}{K}} - \frac{\binom{N^- - 1}{K-1}}{\binom{G-1}{K-1}}\right)\frac{1}{\sigma^{\text{group}}} \\
&= \left(\frac{\prod_{i=0}^{K-1}(N^- - i)}{\prod_{i=0}^{K-1}(N - i)} - \frac{\prod_{i=1}^{K-1}(N^- - i)}{\prod_{i=1}^{K-1}(N - i)}\right)\frac{1}{\sigma^{\text{group}}} \\
&= \left(1 - \frac{G}{N^-}\right)\frac{\prod_{i=0}^{k-1}(N^- - i)}{\prod_{i=0}^{k-1}(G - i)}\frac{1}{\sigma^{\text{group}}} \\
&= -\frac{q}{1-q}A_{\text{pos}}^{@K} \\
&= \left(A_{\text{pos}}^{@K} \cdot \sqrt{\frac{q}{1-q}}\right) \cdot \left(-\sqrt{\frac{q}{1-q}}\right) \\
&= \sqrt{\frac{\binom{N^-}{K}/\binom{G}{K}}{1 - \binom{N^-}{K}/\binom{G}{K}}} \cdot \sqrt{\frac{q}{1-q}}\right) \cdot \hat{A}_{\text{neg}}
\end{aligned}
\tag{12}
$$

By combining Equation (11) and Equation (12), we arrive at Equation (6), completing the derivation.

## D.2 RELATIONSHIP BETWEEN THR AND ENTROPY REGULARIZER

Under some mild assumptions, optimizing THR plays a similar role as regularizing[2] the evolution of the token entropy in a more efficient way. Because, as stated in the main context, THR considers cross-token influence while current analysis on token entropy consider the influence of learning a observing token on itself Cui et al. (2025). We start from Lemma 1 proposed in Cui et al. (2025), which is how the `Cov-KL` regularizer is derived.

**Lemma 1 in Cui et al. (2025):** Let the actor policy $\pi_\theta$ be a tabular softmax policy, the difference of information entropy given states between two consecutive steps satisfy:

$$
\Delta \mathcal{H}^t \triangleq \mathcal{H}(\pi_{\theta(t+1)}) - \mathcal{H}(\pi_{\theta(t)}) = -\text{Cov}_{\boldsymbol{y} \sim \pi_{\theta(t)}(\cdot|x)}\left(\log \pi_{\theta(t)}(\boldsymbol{y} \mid \boldsymbol{x}), \mathbf{l}_y^{t+1} - \mathbf{l}_y^t\right),
\tag{13}
$$

where $\mathbf{l}$ is the logits vector provided by the model after feeding the input $\boldsymbol{x}$. For notational simplicity, we use the superscript $t$ to denote the training step, rather than an exponent. The equation above holds as long as a first-order Taylor expansion is valid at the logits level, independent of the specific model under consideration. In other words, this lemma is agnostic to the mechanism by which $\mathbf{l}$ evolves, which depends on the particular model architecture or parameterization.

Recall the definition of the covariance:

$$
\text{Cov}_{y \sim \pi}(X, Y) = \mathbb{E}_{y \sim \pi}[X \cdot Y] - \mathbb{E}_{y \sim \pi}[X]\mathbb{E}_{y' \sim \pi}[Y].
$$

---

[2]The strength and direction are controlled by the value and sign of hyper-parameter $p$

Equation (13) can then be written as:

$$
\begin{aligned}
\Delta\mathcal{H}^t(\chi) &= -\mathsf{Cov}_{y\sim\pi_{\theta(t)}(\cdot|\chi)}\left(\log\pi_{\theta(t)}(y\mid\chi), \mathbf{l}_y^{t+1}-\mathbf{l}_y^t\right)\\
&= \mathbb{E}_{y\sim\pi_{\theta(t)}}[\log\pi_{\theta(t)}(y\mid\chi)]\mathbb{E}_{y'\sim\pi_{\theta(t)}}[\mathbf{l}_{y'}^{t+1}-\mathbf{l}_{y'}^t] - \mathbb{E}_{y\sim\pi_{\theta(t)}}\left[(\mathbf{l}_y^{t+1}-\mathbf{l}_y^t)\log\pi_{\theta(t)}(y\mid\chi)\right]\\
&= -\mathcal{H}(\pi_{\theta(t)})\mathbb{E}_{y\sim\pi_{\theta(t)}}[\mathbf{l}_y^{t+1}-\mathbf{l}_y^t] - \mathbb{E}_{y\sim\pi_{\theta(t)}}\left[(\mathbf{l}_y^{t+1}-\mathbf{l}_y^t)\log\pi_{\theta(t)}(y\mid\chi)\right]\\
&= -\mathcal{H}(\pi_{\theta(t)})\sum_{v=1}^V\pi_{\theta(t)}(y=v\mid\chi)(\mathbf{l}_v^{t+1}-\mathbf{l}_v^t)-\\
&\qquad \sum_{v=1}^V\pi_{\theta(t)}(y=v\mid\chi)(\mathbf{l}_v^{t+1}-\mathbf{l}_v^t)\log\pi_{\theta(t)}(y=v\mid\chi)\\
&= -\sum_{v=1}^V\pi_{\theta(t)}(y=v\mid\chi)(\mathbf{l}_v^{t+1}+\mathbf{l}_v^t)\left(\mathcal{H}(\pi_{\theta(t)})+\log\pi_{\theta(t)}(y=v\mid\chi)\right)\\
&= -\left\langle\mathcal{H}(\pi_{\theta(t)})\pi_{\theta(t)}(\cdot\mid\chi)+\pi_{\theta(t)}(\cdot\mid\chi)\odot\log\pi_{\theta(t)}(\cdot\mid\chi), \mathbf{l}^{t+1}-\mathbf{l}^t\right\rangle\\
&= -\mathcal{H}(\pi_{\theta(t)})\left\langle\pi_{\theta(t)}(\cdot\mid\chi)+\underbrace{\frac{1}{\mathcal{H}(\pi_{\theta(t)})}\pi_{\theta(t)}(\cdot\mid x)\odot\log\pi_{\theta(t)}(\cdot\mid x)}_{V\times 1,\text{defined as }Q(\chi)}, \mathbf{l}^{t+1}-\mathbf{l}^t\right\rangle\\
&= c\left\langle -Q(\chi)-\pi_{\theta(t)}(\cdot\mid\chi), \mathbf{l}^{t+1}(\chi)-\mathbf{l}^t(\chi).\right\rangle
\end{aligned}
\tag{14}
$$

where the operator $\odot$ is the element-wise multiplication of two vectors, $\chi\triangleq\boldsymbol{x},\boldsymbol{y}_{<k}$ is the context for the prediction of the $k$-th token, and $c$ is a constant for notation conciseness. In the last equation, we reintroduce the input $\chi$ to the notation to remind readers that the entire equation is conditioned on a given context sequence $\chi$. That is an important extension, because most existing works on entropy regularization (e.g., Cui et al. (2025)) **only focus on the influence introduced by updating the observing token on itself**. In other words, the $\chi$ for $Q$ and $l$ are identical. The Cov-KL method compared in Table 4 just applies the quantity above to select tokens with high covariances, and then uses the KL penalty to restrict the update of them.

We here connect THR to entropy in a more systematic way by showing that THR can control the rate of entropy growth $\mathcal{H}^t(\chi)$ through the choice of $p$. Beyond the simplified tabular softmax setting, our analysis extends to more realistic models with shared parameters across tokens. In this case, THR naturally captures the **cross-token** influences that arise throughout the learning process. In other words, when tracking the confidence change of $\pi_{\theta(t)}(y\mid\chi)$, THR accounts for the learning dynamics of all other tokens across all responses, i.e., $\boldsymbol{y}_{i,<k}$ for varying $i$ and $k$.

To make the notations concise, we follow the settings in Ren & Sutherland (2025) and use $\chi_o$ and $\chi_u$ to denote the "observing" token and "updating" context, respectively. Then, Equation (14) becomes:

$$
\Delta\mathcal{H}^t(\chi_o) = c\left\langle -Q(\chi_o)-\pi_{\theta(t)}(\cdot\mid\chi_o), \mathbf{l}^{t+1}(\chi_o)-\mathbf{l}^t(\chi_o)\right\rangle.
$$

Following Deng et al. (2025), and under the unconstrained features assumption Deng et al. (2025); Mixon et al. (2022), we then represent $\mathbf{l}^t(\chi_o)=\mathbf{W}^t\mathbf{h}_o$, where $\mathbf{W}\in\mathbb{R}^{V\times d}$ denotes the shared read-out layer and $\mathbf{h}_o\in\mathbb{R}^{d\times 1}$ is the feature vector produced by the LLM backbone, conditioned on the context sequence $\chi_{u/o}=\boldsymbol{x},\boldsymbol{y}_{u/o,<k}$. Note that while $\mathbf{l}^t(\chi_o)$ shares the same $\mathbf{W}^t$, the feature vector $\mathbf{h}$ differs across contexts due to variations in input sequences. The difference vector $\mathbf{l}^{t+1}(\chi_o)-\mathbf{l}^t(\chi_o)\in\mathbb{R}^{V\times 1}$ can then be expressed as:

$$
\mathbf{l}^{t+1}(\chi_o)-\mathbf{l}^t(\chi_o) = (\mathbf{W}^{t+1}-\mathbf{W}^t)\mathbf{h}_o = -\eta\nabla_{\mathbf{W}}\mathcal{L}(\sigma(\mathbf{W}\mathbf{h}_u), \boldsymbol{e}_u)\mathbf{h}_o,
$$

where $\eta$ is the learning rate, $\sigma(\cdot)$ is the softmax function, and $\boldsymbol{e}_u$ is the one-hot distribution determined by the label of $y_u$. When the cross-entropy loss is considered, the equation above can be simplified to

$$
\mathbf{l}^{t+1}(\chi_o)-\mathbf{l}^t(\chi_o) = \underbrace{(\boldsymbol{e}_u-\pi_{\theta(t)}(\cdot\mid\chi_u))}_{V\times 1}\cdot\underbrace{\mathbf{h}_u^\top\mathbf{h}_o}_{1\times 1}.
$$

Substituting this back to Equation (14), we can get

$$
\Delta\mathcal{H}^t(\chi_o) = c\left\langle -Q(\chi_o)-\pi_{\theta(t)}(\cdot\mid\chi_o), \boldsymbol{e}_u-\pi_{\theta(t)}(\cdot\mid\chi_u)\right\rangle\cdot\mathbf{h}_u^\top\mathbf{h}_o
\tag{15}
$$

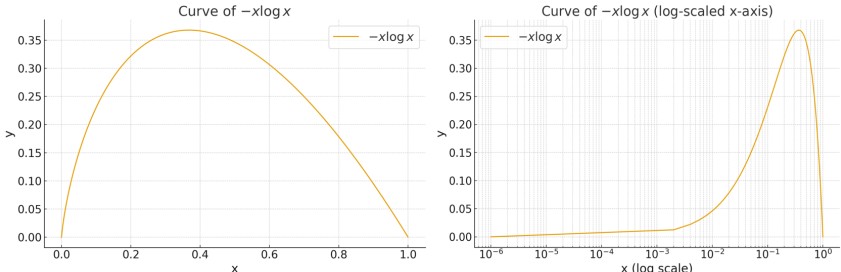

Figure 8: The shape of $-x \log x$ for $x \in (0, 1)$, shown in both the original and logarithmic scales.

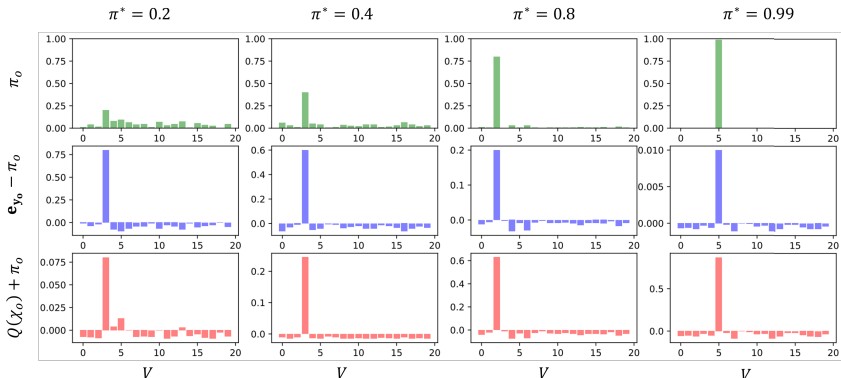

Figure 9: Four examples of the distribution of $\pi$, $\mathbf{e}_o - \pi$ and $Q + \pi$.

Now, recall our definition of THR in Definition 4.1, where for each $k$ in the summation, the term has the format $\langle \mathbf{h}_{\mathbf{x}, \boldsymbol{y}^+_{i,<k}}, \mathbf{h}_{\mathbf{x}, \boldsymbol{y}_{<k'}} \rangle$, which is just $\mathbf{h}_u^\top \mathbf{h}_o$ above. Combining the definition of $\alpha$ and using the notations in this section, we can rewrite the signed-THR as follows:

$$\mathsf{sign}(\boldsymbol{y}_u) \cdot \mathrm{THR}(\boldsymbol{y}_o, \boldsymbol{y}_u, k) = \sum_u \langle \boldsymbol{e}_o - \pi_{\theta(t)}(\cdot \mid \chi_o), \boldsymbol{e}_u - \pi_{\theta(t)}(\cdot \mid \chi_u) \rangle \cdot \mathbf{h}_u^\top \mathbf{h}_o, \qquad (16)$$

where $\mathsf{sign}(\boldsymbol{y}_u)$ depends on whether the completion is correct or not. Now, comparing the inner product in Equation (15) and Equation (16), it is clear that the directional similarity between $-Q(\chi_o)$ and $\boldsymbol{e}_o$ determines the effect introduced by THR and the entropy regularizer.

We now show that, under mild assumptions (which typically hold during LLM fine-tuning), $-Q(\chi_o)$ and $\boldsymbol{e}_o$ point to a very similar direction (measured by their cosine similarity).

This observation follows from the shape of the function $-x \log x$, illustrated in Fig. 8. In a distribution where most probability mass is concentrated on few dimensions, the dominant entry of $\pi_{\theta(t)}^t(\cdot \mid \chi_o) \odot \log \pi_{\theta(t)}^t(\cdot \mid \chi_o)$ is significantly larger than the rest. To validate this, we randomly generate distributions and compute the cosine similarity between $-Q(\chi_o)$ and $\boldsymbol{e}_o$ in Fig. 9 and Fig. 10. The results show a clear trend: as both the vocabulary size and the peakiness of the distribution increase, the alignment between the two vectors becomes stronger.

We now examine the relationship between THR and entropy. Recall that THR is defined as

$$\hat{A}_{i,k}^{\mathrm{THR(p)}} = \mathbb{1}[|\mathrm{THR}_{i,k}| > \tau] \cdot (1 + \mathsf{sign}(\mathrm{THR}_{i,k}) \cdot p) \cdot \hat{A}_{i,k}.$$

When $p < 0$, tokens with larger THR values receive stronger penalties. Since, in most cases, $\Delta \mathcal{H}^t(\chi)$ and THR point in similar directions, this implies that tokens with higher potential entropy change are penalized, closely aligning with the intuition behind Cov-KL. However, as shown in our experiments, THR achieves greater improvements in exploration performance because it explicitly accounts for **cross-token** influence, rather than relying solely on entropy-based signals on a token's self-influence, as in COV-KL Cui et al. (2025).

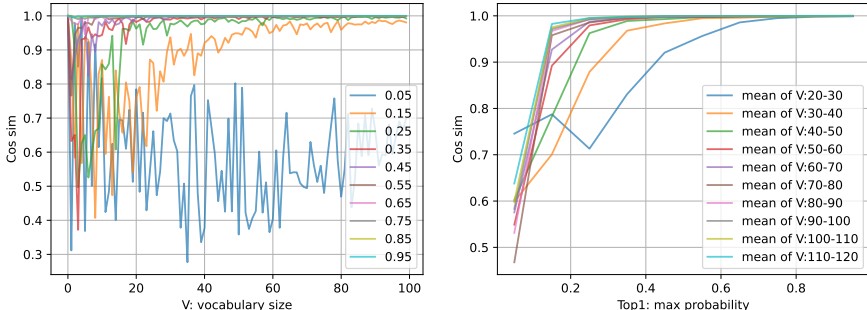

Figure 10: We sweep the value of vocabulary size $V$ and argmax probability of the distribution $\pi^*$. The distribution is generated by fixing $\pi^*$ and randomly assign the extra probability mass to other dimensions. The results show that the cosine similarity between $\mathbf{e}_o - \pi$ and $Q + \pi$ is indeed very large when $V$ and $\pi^*$ are large enough.

# E MORE STUDIES

## E.1 ABLATION STUDY ON $p$

In this section, we conduct ablation study on $p$.

**Ablation Study on $p > 0$ for exploitation:** For exploitation, we evaluated $p \in \{0, 0.05, 0.1, 0.2\}$. The results in Table 11 show that decreasing $p$ from 0.1 to 0.05 achieves the higher greedy accuracy, outperforming GRPO by 2.8%. This suggests that a milder exploitation strength is more suitable for the Qwen2.5-Math-1.5B model. In contrast, increasing $p$ to 0.2 leads to a slight drop in greedy accuracy compared with $p = 0.1$, likely due to excessive exploitation.

| Base Model | Method | Hard Datasets | | | | Standard Datasets | | | | Total Avg. |
|---|---|---|---|---|---|---|---|---|---|---|
| | | AIME25 | AIME24 | AMC23 | Hard Avg. | MATH500 | Minerva | Olympiad | Standard Avg. | |
| | Base | 0.0 | 3.3 | 20.0 | 7.8 | 39.6 | 7.7 | 24.9 | 24.1 | 15.9 |
| | GRPO | 3.3 | **13.3** | 57.5 | 24.7 | **71.8** | 29.0 | 34.1 | 45.0 | 34.8 |
| Qwen2.5-Math-1.5B | THR | 3.3 | **13.3** | 55.0 | 23.9 | 70.8 | 32.4 | 34.1 | 45.8 | 34.8 |
| | THR ($p = -0.1$) | **10.0** | **13.3** | 60.0 | 27.8 | 70.6 | 32.0 | 32.7 | 45.1 | 36.4 |
| | THR ($p = 0.05$) | **10.0** | **13.3** | **62.5** | **28.6** | **71.8** | **35.7** | 32.1 | **46.5** | **37.6** |
| | THR ($p = 0.1$) | 3.3 | **13.3** | **62.5** | 26.4 | 71.4 | 33.1 | **34.5** | 46.3 | 36.3 |
| | THR ($p = 0.2$) | 3.3 | **13.3** | 60.0 | 25.5 | 71.0 | 32.7 | 33.9 | 45.9 | 35.7 |

Table 11: Exploitation Results on hard and standard math datasets. Pass@1 accuracy (%) using greedy decoding across different methods and datasets. **Bold** is best performance, underline is second-best.

**Ablation Study on $p < 0$ for exploration.** For exploration, we evaluate $p \in \{0, -0.05, -0.1, -0.2\}$. As shown in Table 12, we observe a consistent exploration trend where all three $p$ can consistently improve the pass@K performance over GRPO, thus reinforcing the conclusion that $p < 0$ can enhance exploration.

## E.2 GRADIENT STEPS AND CONVERGENCE

**Effective Gradient Steps.** We note that a "step" in our setup corresponds to 32 gradient steps. We follow standard GRPO practice and with a prompt batch size of 256 and 8 rollouts per prompt. Then we use a mini-batch size of 64, resulting in 32 gradient steps per step. Therefore, 40 steps corresponds to 1280 gradient steps.

**Validation accuracy along Steps.** We show the convergence of training by demonstrating the accuracy of validation dataset of MATH (levels 3–5) Hendrycks et al. (2021), as shown in Figure 11, the validation performance continues to improve gradually until around 30-35 steps, after which the increasing is flat, indicating that the model is convergence, thus we use 40 steps for consistency.

| Method | Qwen2.5-math-1.5B Pass@K | | | | | | | | |
|---|---|---|---|---|---|---|---|---|---|
| | 1 | 2 | 4 | 8 | 16 | 32 | 64 | 128 | 256 |
| **AIME 2025** | | | | | | | | | |
| GRPO | 5.9 | 9.9 | 15.0 | 20.5 | 26.5 | 33.6 | 41.5 | 49.8 | 56.7 |
| THR | 5.4 | 9.2 | 14.1 | 19.4 | 25.0 | 31.7 | 39.5 | 48.0 | 56.7 |
| THR ($p = -0.05$) | 5.9 | 10.1 | 15.5 | 21.2 | 27.5 | 34.7 | 42.0 | 49.6 | 60.0 |
| THR ($p = -0.1$) | **6.0** | 10.1 | 15.3 | 20.9 | 26.8 | 33.9 | 41.7 | **50.0** | **60.0** |
| THR ($p = -0.2$) | **6.0** | **10.2** | **15.6** | **21.4** | **28.1** | **36.2** | **44.0** | 49.8 | 53.3 |
| **AIME 2024** | | | | | | | | | |
| GRPO | 11.4 | 17.7 | 24.3 | 30.5 | 36.7 | 43.4 | 50.0 | 56.0 | 63.3 |
| THR | 10.6 | 16.7 | 23.4 | 30.2 | 37.2 | 44.8 | 51.9 | 58.5 | 63.3 |
| THR ($p = -0.05$) | 11.9 | 18.2 | 24.8 | 31.0 | 37.5 | 44.8 | 52.9 | 61.6 | 70.0 |
| THR ($p = -0.1$) | 11.9 | 18.2 | **24.9** | **31.2** | 37.9 | 45.3 | 52.9 | 61.2 | **70.0** |
| THR ($p = -0.2$) | **12.2** | **18.3** | 24.7 | **31.2** | **38.6** | **47.4** | **56.6** | **64.0** | **70.0** |
| **AMC 2023** | | | | | | | | | |
| GRPO | 46.6 | 59.1 | 70.0 | 78.9 | 85.5 | 90.2 | 93.7 | 96.0 | 97.5 |
| THR | 44.8 | 57.8 | 69.1 | 78.2 | 85.1 | 90.1 | 93.6 | 95.9 | 97.5 |
| THR ($p = -0.05$) | 48.1 | 60.6 | 71.2 | 79.5 | 85.3 | 89.8 | 93.6 | 97.1 | **100.0** |
| THR ($p = -0.1$) | 47.9 | 61.0 | 72.2 | **81.1** | **87.3** | **91.6** | 95.1 | 98.0 | **100.0** |
| THR ($p = -0.2$) | **50.3** | **62.4** | **72.4** | 80.4 | 86.4 | 91.3 | **95.4** | **98.3** | **100.0** |
| **Average** | | | | | | | | | |
| GRPO | 21.3 | 28.9 | 36.4 | 43.3 | 49.6 | 55.7 | 61.7 | 67.3 | 72.5 |
| THR | 20.3 | 28.0 | 35.5 | 42.6 | 49.1 | 55.5 | 61.7 | 67.5 | 72.5 |
| THR ($p = -0.05$) | 22.0 | 29.6 | 37.2 | 43.9 | 50.1 | 56.4 | 62.8 | 69.4 | 76.7 |
| THR ($p = -0.1$) | 21.9 | 29.8 | 37.5 | **44.4** | 50.7 | 57.3 | 63.2 | 69.7 | **76.7** |
| THR ($p = -0.2$) | **22.8** | **30.3** | **37.6** | 44.3 | **51.0** | **58.3** | **65.3** | **70.7** | 74.4 |

Table 12: Pass@K performance of different $p < 0$ for Qwen2.5-math-1.5B.

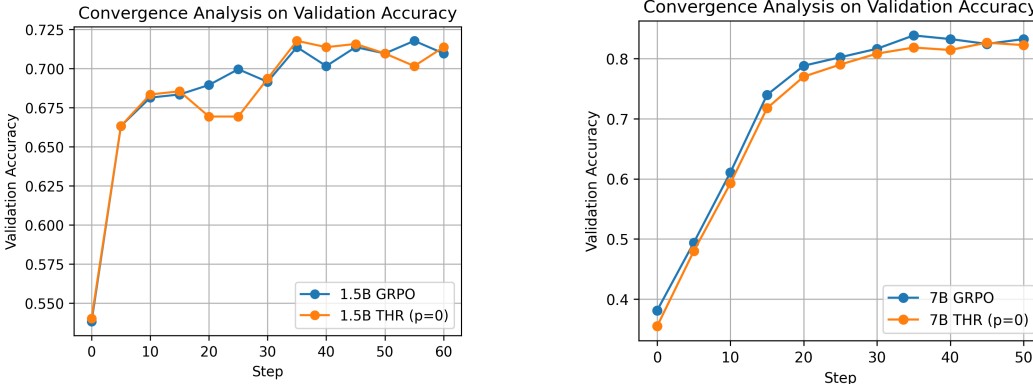

Figure 11: Validation accuracy along training of Qwen2.5-Math-1.5B and Qwen2.5-Math-7B

**Reward along Steps.** For completeness, we include the reward curves in Figure 12. As shown, the reward rises during the early phase and then stabilizes around 0.55 for the 1.5B model and 0.6 for the 7B model, demonstrating that training remains stable throughout. Although dynamic filtering prevents the reported reward from capturing the true correctness of model outputs, it remains a useful proxy for assessing training stability.

### E.3 COMPARISON WITH CLIP-HIGH

In this section, we compare against the clip-high baseline Yu et al. (2025) using the recommended clipping value of 0.28. As shown in Table 13, clip-high improves exploration for $K \geq 32$ relative to GRPO. Nevertheless, despite its strength, THR ($p < 0$) consistently surpasses clip-high across all $K$, highlighting the effectiveness of THR ($p < 0$) in enhancing exploration.

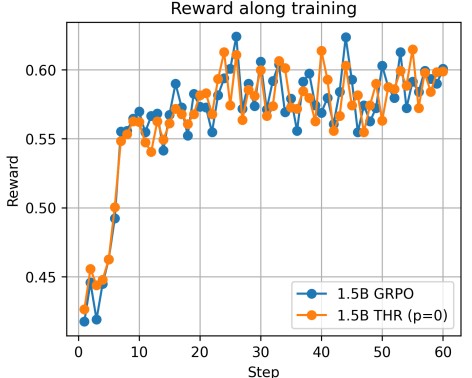 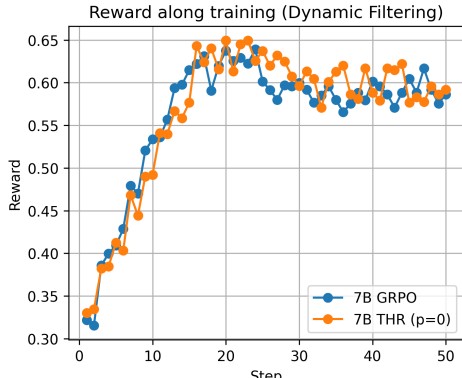

Figure 12: Reward (dynamic filtering applied) along training of Qwen2.5-Math-1.5B and Qwen2.5-Math-7B.

| Method | Qwen2.5-Math-1.5B Pass@K | | | | | | | | |
|---|---|---|---|---|---|---|---|---|---|
| | 1 | 2 | 4 | 8 | 16 | 32 | 64 | 128 | 256 |
| **AIME 2025** | | | | | | | | | |
| GRPO | 5.9 | 9.9 | 15.0 | 20.5 | 26.5 | 33.6 | 41.5 | 49.8 | 56.7 |
| Clip-High | 5.6 | 9.6 | 14.8 | 20.5 | 26.6 | 33.7 | 41.6 | 48.4 | 53.3 |
| THR ($p < 0$) | **6.0** | **10.1** | **15.3** | **20.9** | **26.8** | **33.9** | **41.7** | **50.0** | **60.0** |
| **AIME 2024** | | | | | | | | | |
| GRPO | 11.4 | 17.7 | 24.3 | 30.5 | 36.7 | 43.4 | 50.0 | 56.0 | 63.3 |
| Clip-High | 10.8 | 16.7 | 23.2 | 29.8 | 36.5 | 44.0 | 52.1 | 60.7 | 70.0 |
| THR ($p < 0$) | **11.9** | **18.2** | **24.9** | **31.2** | **37.9** | **45.3** | **52.9** | **61.2** | **70.0** |
| **AMC23** | | | | | | | | | |
| GRPO | 46.6 | 59.1 | 70.0 | 78.9 | 85.5 | 90.2 | 93.7 | 96.0 | 97.5 |
| Clip-High | 47.3 | 59.9 | 70.5 | 78.8 | 84.9 | 89.8 | 93.8 | 97.3 | 100.0 |
| THR ($p < 0$) | **47.9** | **61.0** | **72.2** | **81.1** | **87.3** | **91.6** | **95.1** | **98.0** | **100.0** |
| **Average** | | | | | | | | | |
| GRPO | 21.3 | 28.9 | 36.4 | 43.3 | 49.6 | 55.7 | 61.7 | 67.3 | 72.5 |
| Clip-High | 21.2 | 28.7 | 36.2 | 43.0 | 49.3 | _55.8_ | _62.5_ | _68.8_ | _74.4_ |
| THR ($p < 0$) | **21.9** | **29.8** | **37.5** | **44.4** | **50.7** | **57.3** | **63.2** | **69.7** | **76.7** |

Table 13: Comparing exploration ability with Pass@$K$ on Qwen2.5-Math-1.5B across AIME 2024, AIME 2025, and AMC23.

### E.4 STUDY ON ERROR-CORRECTION BEHAVIOR.

In this section, we investigate how THR ($p < 0$) relates to corrective and self-verifying behaviors. To quantify this, we compute the ratio of reflection-related tokens to the total number of generated tokens. The full list of reflection-related words used for this analysis is provided in Table 14.

| Reflection Words | | | |
|---|---|---|---|
| actually | although | alternating | but |
| correct | despite | error | fix |
| however | incorrect | instead | mistake |
| nevertheless | nonetheless | note | realize |
| realized | rethink | reconsider | still |
| thinking | think | though | wait |
| whereas | otherwise | wrong | yet |
| unless | | | |

Table 14: List of reflection-related words.

| Qwen2.5-Math-1.5B | |
|---|---|
| **Method** | **#Reflection Token / #Token** |
| GRPO | 0.34% |
| THR | 0.36% |
| THR ($p = -0.1$) | 0.55% |

Table 15: #Reflection Token / #Token ratio for Qwen2.5-Math-1.5B.

We then report the frequency of these tokens in Table 15, which shows that setting $p < 0$ increases the presence of reflection tokens. This indicates that THR ($p < 0$) can encourage more verification and correction behavior.

### E.5 Tokens Retained

The threshold $\tau$ is inherently adaptive, as it is defined as the average influence of a correct response's token on the likelihoods of all correct responses. We report in Table 16 the average proportion of tokens retained under this threshold for Qwen2.5-Math-1.5B and Qwen2.5-Math-7B. Notably, the 7B model retains fewer high-THR tokens, which is expected: a stronger model possesses more knowledge, is more confident in its answers, and therefore relies on fewer influential tokens.

| Model | Avg. Ratio Retained |
|---|---|
| Qwen2.5-Math-1.5B | 18% |
| Qwen2.5-Math-7B | 14% |

Table 16: Average fraction of tokens retained under the adaptive threshold $\tau$.

## F Usage of Large Language Model

In preparing this paper, we made limited use of ChatGPT to support writing and editing. Specifically, LLMs were employed for language polishing, grammar refinement, and rephrasing sentences to improve clarity and readability. Importantly, all technical content, including theoretical analysis, algorithm design, and experimental results, was conceived, implemented, and validated by the authors. LLM outputs were always critically reviewed, verified, and revised before inclusion. No LLM-generated text, figures, or tables were incorporated without careful human oversight.

