# OpenReview forum: "Token Hidden Reward: Steering Exploration-Exploitation in Group Relative Deep Reinforcement Learning"
_ICLR.cc/2026/Conference — ICLR 2026 Poster_

### Official Review · Reviewer_on7i · 2025-10-29

**Soundness:** 3
**Presentation:** 2
**Contribution:** 2
**Rating:** 6
**Confidence:** 2

**Summary:**

This paper introduces Token Hidden Reward (THR), a token-level metric that quantifies how each generated token contributes to the change in the likelihood of correct responses under GRPO-style RL with verifiable rewards. The key idea is to decompose the learning dynamics into per-token “hidden rewards” whose magnitude identifies a small set of dominant tokens driving updates, and whose sign aligns with the exploration–exploitation trade-off: positive THR tends to strengthen confidence in correct outputs (exploitation), while negative THR tends to preserve probability mass for alternative outputs (exploration). Building on this insight, the authors propose a simple token-level advantage reweighting scheme that:

- masks low-influence tokens (dominant-token training),
- and amplifies tokens by sign to bias toward exploitation (p>0) or exploration (p<0).

**Strengths:**

Originality

- A clear, token-level decomposition of learning dynamics under GRPO that cleanly ties per-token influence to exploration vs. exploitation via the sign of THR. This is a novel perspective beyond question-level hardness reweighting or per-token self-entropy methods, and extends prior analysis (e.g., negative-gradient focus) to both signs and cross-token effects

- The framing of “dominant tokens” and the empirical finding that masking to high-|THR| tokens retains performance is a crisp, interpretable contribution that could inform more efficient training

Quality
- Theoretical grounding connects THR to likelihood dynamics and relates it to entropy regularization; the analysis explains why the sign of THR modulates exploration–exploitation

- Broad experimental sweep across models (Qwen 0.5B/1.5B/7B; Llama3.2-3B) and RL objectives (GRPO, GSPO-token). The p>0 vs p<0 behavior is consistently demonstrated; THR compares favorably to Pass@K-mixed and COV-KL on Pass@K

- Practicality: THR’s compute overhead is small relative to data generation, suggesting the technique scales. The paper explicitly reports module-wise runtimes, which is valuable for practitioners

Clarity

- Clear definition of THR and reweighting, with equations and intuitive illustrations (e.g., sign interpretation, density plots of THR, overlap with high-entropy tokens). The narrative links theory → algorithm → empirical behavior in a readable way

Significance

- A fine-grained knob to trade-off greedy accuracy vs. Pass@K with minimal engineering changes can be valuable in RLVR pipelines, where one may want different behaviors by domain or deployment constraints (e.g., high-confidence single-shot vs. BoN sampling)

- The “dominant tokens” lens might inspire further token-level curricula, diagnostics, or adaptive schedules.
Additionally reflecting the user’s perspective: the method is appealing because computing THR is relatively lightweight compared to rollouts, making it amenable to scaling and combination with existing RLVR loops.

**Weaknesses:**

Magnitude and consistency of gains

- While some settings show non-trivial improvements (e.g., up to ~4 points Pass@1 on Qwen-7B), in several cases improvements over strong baselines like GRPO are modest and sometimes within typical training variance bands, especially at smaller model sizes or across certain datasets. Emphasize variance estimates (e.g., CI/error bars across seeds) to quantify significance of the deltas

- The exploration benefits at large K are clear, but the practical importance of very large K (e.g., 128–256) may be less relevant for many real-world budgets. Provide a sharper focus on K in the 4–16 range and on success-vs-cost trade-offs

Generality beyond verifiable domains

- The method and analysis are anchored in RLVR with binary/verifiable rewards and GRPO/GSPO-style group structures. It remains unclear how to port THR to non-verifiable domains (e.g., preference models, open-ended generation) where correctness signals are noisy/subjective. The paper mentions generalization across RL objectives but not across reward types or unverifiable tasks

Interaction with error-correction behavior
- The paper interprets negative THR as “preserving probability mass for alternative (than the correct) responses” to encourage exploration. It is less clear how this interacts with error-correction/repair behaviors, where one typically wants to actively down-weight error-inducing trajectories and prioritize corrective steps. More analysis is needed on whether amplifying negative-THR tokens inadvertently reinforces patterns that impede self-correction or verifier-guided repair

Scope of analysis and ablations

- Threshold τ selection follows a prior influence-based heuristic. Sensitivity analyses on τ, the fraction of tokens retained, and p schedules are limited in the main text. Given the centrality of these choices, ablate:
absolute vs. percentile thresholds for |THR|,
dynamic τ over training, adaptive p tied to q (group accuracy) or per-question difficulty

- Baselines could include token-level reweighting heuristics that don’t require THR (e.g., top-|grad| tokens, per-token loss magnitude, or simple entropy-top-k), to isolate THR’s unique value beyond “focus on hard tokens.” Some overlap with entropy is discussed, but stronger head-to-head controls would increase confidence

Reporting and diagnostics

- The paper makes strong claims about cross-token interactions; more direct diagnostics (e.g., intervention studies that swap token subsets, or causal tests across positions) would bolster this claim beyond correlation/overlap with entropy

- Calibration and stability analyses are missing. Since THR aims to shape confidence, report calibration metrics, entropy dynamics over training, and instabilities (e.g., length collapse, variance across seeds) more prominently for each setting

**Questions:**

Interesting work, I wonder how this ties to goal-conditioned RL and if there some applications there.

---

> ### Author Response · Authors · 2025-11-21
> **Thank you very much! (part 1)**
>
> Thank you for your thorough evaluation and helpful suggestions. We truly appreciate your time and address your comments below.
>
> ----------- W1: Magnitude of gains--------
>
> **Answer:**   Thank you for your question. We respectfully disagree that some improvements are due to randomness. The gains we observe are substantial in magnitude, consistent across model scales and architectures, and far exceed typical variance levels, indicating that they are not attributable to stochastic noise.
>
> - Deterministic and low-variance evaluation. Greedy decoding accuracy is deterministic, and Pass@K scores are computed from 256 independent rollouts, ensuring stable and low-variance estimates.
>
> - Magnitude of improvements. For exploitation, Qwen2.5-Math-7B with p>0 improves greedy accuracy by +4.1% over GRPO and +3.5% over THR(p=0). On smaller models (0.5B and 1.5B), the gains remain consistent, exceeding +1.5%. For exploration, the improvement is also pronounced: on Qwen2.5-Math-1.5B, Pass@K increases by up to +4%, and on Llama3.2-3B, THR(p<0) achieves roughly +10% improvement over GRPO at K=256. These are not minor fluctuations but clear, directional performance gains.
>
> - Directional consistency across all K and scales indicates a robust effect, not noise. THR(p<0) consistently surpasses both GRPO and THR(p=0) across all K values, typically by more than 1 % in all K. This monotonic trend across Qwen 0.5B–7B and Llama 3B demonstrates that the observed effects are systematic and scale-consistent, rather than random.
>
> - Importantly, the primary aim of this work is to show that a simple adjustment of p reliably steers the model toward exploration or exploitation; we achieve these improvements without careful hyperparameter tuning. To further support this, we conducted an ablation on Qwen2.5-Math-1.5B. We found that p=0.05 improves greedy accuracy by +2.8 % over GRPO, and using p=−0.2 improves Pass@32–128 by around +3 %. These results reinforce that the observed gains are robust and directionally consistent.
>
> We thank the reviewer’s question and will add the discussion and clarify these points in the revised version.
>
> ----------- W1:  The exploration benefits at large K are clear, but the practical importance of very large K (e.g., 128–256) may be less relevant for many real-world budgets.--------
>
> **Answer:** Thank you for the comment. We would first like to emphasize that the primary aim of this work is to show that a simple adjustment of p, THR(p<0) consistently surpasses both GRPO and THR(p=0) across all K values, typically by more than 1 % in all K. We also conducted an ablation on Qwen2.5-Math-1.5B, using p=−0.2  improves Pass@32–128 by around +3 %.
>
> At the same time, evaluating larger K remains highly important:
>
> - Test-time scaling requires Large K. Recent work on test-time compute scaling [1] shows that allowing a model to use a fixed but non-trivial amount of inference-time computation can dramatically improve performance, sometimes enabling a much smaller model to outperform a model 14× larger. In this setting, systems routinely allocate hundreds of rollouts per query [2, 3], ensuring the motivation of a larger K.
>
> - Large-K in training regimes. Many real-world training pipelines use large sampling budgets. For example, AlphaCode reports generating millions of samples per problem to find a single correct program [4]. Such regimes rely heavily on exploration on the larger K.
>
> [1] Scaling LLM test-time compute optimally can be more effective than scaling model parameters.
>
> [2]  Deep think with confidence.
>
> [3] Self-consistency improves chain-of-thought reasoning in language models.
>
> [4] Competition-level code generation with alphacode.

---

> ### Author Response · Authors · 2025-11-21
> **Thank you very much! (part 2)**
>
> ------- W2: beyond verifiable domains, how to port THR to non-verifiable domains ------
>
> **Answer:** Thank you for the suggestion. Extending THR beyond verifiable domains is indeed an interesting direction, and we can see that THR has strong potential in such settings. Although our main analysis focuses on RLVR-style rewards, the core idea behind THR, measuring token-level influence on relative preference signals, does not require the reward to be strictly verifiable. As long as we can determine those preferred responses.
> To illustrate this, we conducted a preliminary study using offline DPO, since it provides an efficient learning setup. We use GSM8K and deliberately recast the task as a preference problem by:
> - Using the ground-truth reasoning and answer as the preferred response.
> - Fine-tuning Llama-3-8B on GSM8K to make the preferred ground-truth reasoning more online and generate 4 candidate responses for each question using the finetuned model.
> - Selecting the dispreferred samples using two methods: THR-based selection: choosing the candidate with the lowest sum of THR across all tokens, and Random selection: a baseline negative example.
>
> We then apply DPO using these selected negative samples and the reference answer as preferred.
>
> | Method | Accuracy |
> |--------|----------|
> | Finetuned baseline | 44.88 |
> | Finetuned + DPO (THR-selected negatives) | 47.76 |
> | Finetuned + DPO (random negatives) | 45.71 |
>
> The results above show that THR provides a more informative and effective negative-sample signal than random selection, even when correctness is framed as a preference rather than a binary verifiable reward. While fully extending THR to online GRPO in open-ended, unverifiable tasks is outside the scope of this paper, our initial findings demonstrate promising potential. We regard this direction as future work
>
> ---------- W3: Interaction with error-correction behavior ------
>
> **Answer:** Thank you for your thoughtful question. We believe there may be a misunderstanding about the role of negative THR in our framework.  First, as stated in lines 201–204 of the paper, we interpret setting p<0 as producing a smaller increase in the likelihood of the observed correct responses, thereby preserving probability mass for other potentially correct alternatives. This encourages exploration rather than reinforcing incorrect behaviors. Second, tokens with negative THR are not equivalent to tokens from incorrect trajectories, as shown in Fig. 2, both correct and incorrect responses contain negative-THR tokens.
>
> Finally, following your suggestion, we examined whether THR (p<0) inadvertently suppresses corrective or self-verifying behavior. Specifically, we measured the frequency of reflection-related tokens. We define reflection-related tokens as:
>
> | **Reflection Words** |  |  |  |
> |----------------------|----|----|----|
> | actually     | although    | alternating | but |
> | correct      | despite     | error       | fix |
> | however      | incorrect   | instead     | mistake |
> | nevertheless | nonetheless | note        | realize |
> | realized     | rethink     | reconsider  | still |
> | thinking     | think       | though      | wait |
> | whereas      | otherwise   | wrong       | yet |
> | unless       |             |             |     |
>
> We report the frequency of such tokens of Qwen2.5-Math-1.5B in the Table below,  showing that setting p<0 actually increases the occurrence of these reflection tokens:
>
> | Method      | #reflection token / #token |
> |-------------|-----------------------------|
> | GRPO        | 0.34% |
> | THR         | 0.36% |
> | THR ($p=-0.1$) | **0.55%** |
>
> This suggests that THR with p<0 instead amplifies the emergence of reflection tokens.  In short, p<0 and reflection are two causally related concepts, focusing on different levels of the model: manipulating THR improves exploration ability, hence arousing reflection behaviors.
>
> ------- W4: Scope of analysis and ablations -------
>
> **P1**:  Adaptive threshold t and tokens retained.
>
> **Answer:** The threshold τ is inherently adaptive, as it is defined as the average influence of a correct response’s token on the likelihoods of all correct responses. Following your suggestion, we report below the average proportion of tokens retained under this threshold for Qwen2.5-Math-1.5B and Qwen2.5-Math-7B. Notably, the 7B model retains fewer high-THR tokens, which is expected: a stronger model has more knowledge, thus is more confident in answering given questions and thus drives by fewer tokens.
>
> | Model | Average Ratio Retained |
> |--------|------------------------|
> | 1.5B   | 18% |
> | 7B     | 14% |

---

> ### Author Response · Authors · 2025-11-21
> **Thank you very much! (part 3)**
>
> **P2**:  Ablation study on p.
>
> **Answer:**   Following your suggestion, we conducted a sensitivity analysis on p using **Qwen2.5-Math-1.5B**.
>
> (1) *For exploration*, we use p = [0,-0.05,-0.1,-0.2]. We observe a consistent exploration trend as shown in the table below (complete results in Table 12 of the revised appendix ), where setting p<0 can consistently improve the pass@K performance, thus reinforcing the conclusion that p<0 reliably enhances exploration.
>
> | Method | 1 | 2 | 4 | 8 | 16 | 32 | 64 | 128 | 256 |
> |-------|---|---|---|---|----|----|----|-----|-----|
> | **Average** |||||||||||
> | GRPO | 21.3 | 28.9 | 36.4 | 43.3 | 49.6 | 55.7 | 61.7 | 67.3 | 72.5 |
> | THR | 20.3 | 28.0 | 35.5 | 42.6 | 49.1 | 55.5 | 61.7 | 67.5 | 72.5 |
> | THR (p = −0.05) | $\underline{22.0}$ | 29.6 | 37.2 | 43.9 | 50.1 | 56.4 | 62.8 | 69.4 | 76.7 |
> | THR (p = −0.1) | 21.9 | $\underline{29.8}$ | $\underline{37.5}$ | **44.4** | $\underline{50.7}$ | $\underline{57.3}$ | $\underline{63.2}$ | $\underline{69.7}$ | **76.7** |
> | THR (p = −0.2) | **22.8** | **30.3** | **37.6** | $\underline{44.3}$ | **51.0** | **58.3** | **65.3** | **70.7** | 74.4 |
>
> We also examined the effect of p on Llama. As shown in Fig. 5 and Table 9, setting p=−0.1  or −0.2 consistently improves exploration, with p=−0.2 achieving a +10% gain over GRPO at larger K=256. This result highlights that, even though Llama3.2-3B is a non-reasoning model with shorter response lengths (Fig. 6), applying p<0 still reliably enhances exploration.
>
> (2) *For exploitation*, we evaluated p∈[0,0.05,0.1,0.2]. The results show that p=0.05 yields the best greedy accuracy, outperforming GRPO by 2.8% and p<0 by 1.2%, indicating that a milder exploitation strength can be more effective for the weaker qwen2.5-Math-1.5B model. Increasing p to 0.2 slightly decreases greedy accuracy relative to p=0.1,  likely due to over-emphasizing exploitation.
>
> | **Method** | AIME25 | AIME24 | AMC23 | Hard Avg. | MATH500 | Minerva | Olympiad | Standard Avg. | Total Avg. |
> |------------|--------|--------|--------|-----------|---------|---------|----------|----------------|------------|
> | Base | 0.0 | 3.3 | 20.0 | 7.8 | 39.6 | 7.7 | 24.9 | 24.1 | 15.9 |
> | GRPO | 3.3 | **13.3** | 57.5 | 24.7 | **71.8** | 29.0 | $\underline{34.1}$ | 45.0 | 34.8 |
> | THR | 3.3 | **13.3** | 55.0 | 23.9 | 70.8 | 32.4 | $\underline{34.1}$ | 45.8 | 34.8 |
> | THR ($p=-0.1$) | **10.0** | **13.3** | 60.0 | $\underline{27.8}$ | 70.6 | 32.0 | 32.7 | 45.1 | $\underline{36.4}$ |
> | THR ($p=0.05$) | **10.0** | **13.3** | **62.5** | **28.6** | **71.8** | **35.7** | 32.1 | **46.5** | **37.6** |
> | THR ($p=0.1$) | 3.3 | **13.3** | **62.5** | 26.4 | 71.4 | $\underline{33.1}$ | **34.5** | $\underline{46.3}$ | 36.3 |
> | THR ($p=0.2$) | 3.3 | **13.3** | 60.0 | 25.5 | 71.0 | 32.7 | 33.9 | 45.9 | 35.7 |
>
> **Adaptive p:**  Your suggestion to explore adaptive schedules (e.g., linking p to question difficulty or group accuracy q) is highly insightful. As this work focuses on establishing the directional effect of p, we leave adaptive strategies as an exciting direction for future research. Thank you again for the valuable feedback.
>
> At last, we wish to clarify that we didn’t conduct careful hyperparameter tuning, as the goal of this work is to show that a simple adjustment of p can reliably steer the model toward exploration or exploitation.
>
> **P3**:  top-|grad| tokens, per-token loss magnitude, or simple entropy-top-k
>
> **Answer:** Thank you for this question. If we understand correctly, you are referring to Figure 3, where the correlation is between the magnitude (absolute value) of THR and entropy.  However, we emphasize that THR provides signed directional information and the sign steers the exploitation and exploration, whereas methods like top-entropy, per-token loss magnitude heuristics are unsigned and only correlated with THR magnitude.
>
> More specifically, THR-sign indicates whether a token increases or decreases the likelihood of a correct response. This directional information is precisely what enables THR-sign to control exploitation (p>0) versus exploration (p<0). Entropy or per-token loss magnitude alone cannot provide such behavioral control. Thus, although THR and entropy (or others) correlate in magnitude, the directional information provided uniquely by the THR sign is essential for the controlled exploration–exploitation behavior studied in our work.

---

> ### Author Response · Authors · 2025-11-21
> **Thank you very much! (part 4)**
>
> ----------- W4: Reporting and diagnostics ------
>
> **P1:**  cross-token interactions diagnostics
>
> **Answer:** Thank you for your question. The cross-token interaction property of THR follows directly from its definition.   As shown in Definition 4.1, THR measures the change in the log-likelihood of the sum of tokens in response when a single token’s gradient is applied, thus THR inherently captures cross-token influence by construction.
>
> We appreciate the reviewer’s suggestion regarding intervention-style diagnostics. However, swapping tokens or performing causal perturbations often breaks the semantic coherence of reasoning trajectories in RL settings, producing off-distribution responses that destabilize the training. As the theoretical formulation already guarantees that THR encodes cross-token effects, we regard such controlled interventions as an interesting direction for future work.
>
> **P2:**  more stability analysis
>
> **Answer:** To highlight the stability of training, we add the accuracy on the MATH (levels 3–5) validation set over the course of training for Qwen2.5-Math-1.5B and Qwen2.5-Math-7B in Appendix Fig. 11 of the revised version. The validation accuracy increases smoothly and consistently, showing no signs of instability, and gradually plateaus around 30–35 steps, indicating that the models reach a stable convergence point.  We also present the reward curves in Fig. 12 in the appendix (Although dynamic filtering prevents the reported reward from capturing the true correctness of model outputs, it remains a useful proxy for assessing training stability).  As shown, the reward rises steadily during the early phase and then settles into a stable plateau, around 0.55 for the 1.5B model and 0.6 for the 7B model, further confirming that the training process remains consistently stable.
>
> ----------- Q1: ties to goal-conditioned RL ------
>
> **Answer:**  Thank you for the kind words and insightful question. We can see some connection between our framework and goal-conditioned reinforcement learning (GCRL).
>
> In GCRL, agents are trained to reach a specified goal, often by comparing their behavior to reference goal states. Similarly, in our framework, a set of correct reference solutions can be viewed as defining the “goal,” and THR serves as a token-level reward by measuring how each token in a generated response increases or decreases the model’s likelihood of achieving that goal.  Since many goals admit multiple correct solutions, incorporating a diverse set of reference completions allows THR to preserve probability mass across plausible reasoning paths—encouraging exploration. Conversely, using a small set of high-probability solutions focuses learning on narrow patterns, promoting exploitation.
>
> We believe there is a conceptual connection to GCRL, which may be worth exploring in future work.

---

### Official Review · Reviewer_zvLX · 2025-11-01

**Soundness:** 3
**Presentation:** 3
**Contribution:** 3
**Rating:** 6
**Confidence:** 3

**Summary:**

This paper introduces Token Hidden Reward (THR), a token-level metric that quantifies how individual tokens influence the change in the likelihood of correct responses in LLM-RL training. First, the paper analyzes how THR is distributed in correct and incorrect responses and it shows that majority of tokens have 0 THR values and small subset of tokens have large positive and negative values. Based on this insight, the paper proposes two training scenarios with THR, THR-only, where advantages of only tokens with sufficiently large THR values are used, and THR with exploration or exploitation, where reweighting advantages based on THR values. In experiments, THR-only shows comparable performance to the original GRPO variant, which indicates that only small subset of tokens influence training performance. The experiments also show that reweighting advantages based on THR values can steer exploration and exploitation and lead to improved performance in math tasks. Lastly, the paper shows connection between THR and entropy, which indicates that high THR tokens and high entropy tokens are overlapped.

**Strengths:**

The paper is well written and comprehensive. The concept of THR is interesting and generic to LLM-RL settings. And, the nice thing is that the THR values can be computed without additional training components so that it can be essentially used in a plug-and-play style. Also, the paper shows that the THR has an interesting property, which is that small subsets of tokens with large THR values dominate training performance. This fact is a new insight along with how token entropies involve LLM-RL training. The paper provides extensive experiments to show how THR can steer exploration and exploitation and how it can improve performance with math tasks.

**Weaknesses:**

Although I couldn't find obvious flaws in this paper, there some comments that should be addressed:
- How much does THR calculation add computation cost? I imagine that computing THR every time is not a trivial thing to do. The authors should provide information about this.
- This is a fundamental question about the proposal in the paper. Since THR and entropy are high correlated, can we just use the entropy as a proxy of THR? If so, the implementation of algorithms with THR could be simplified. Is there a practical reason to keep using THR over entropies? This seems clarified in the manuscript.

**Questions:**

`Weakness` section includes questions.

---

> ### Author Response · Authors · 2025-11-21
> **Thank you very much!**
>
> Thank you for your thorough evaluation and helpful suggestions. We truly appreciate your time and address your comments below.
>
> ------ W1: computation cost ------
>
> **Answer:** Thank you for your question. The THR computation is actually quite lightweight. As shown in Section C.5 of the appendix (Table 10), we conducted a detailed measurement of the overhead introduced by the THR module. For convenience, we summarize the results below:
>
> | Model | Avg THR Time/Step | Avg Total Time/Step | Ratio |
> |-------|--------------------|----------------------|--------|
> | Qwen2.5-Math-1.5B | 37  | 834  | 0.04 |
> | Qwen2.5-Math-7B   | 39  | 1206 | 0.03 |
> | Llama3.2-3B        | 26  | 968  | 0.03 |
>
> These results show that THR adds only 3–4% additional time per training step, which is small compared to the overall GRPO computation.
>
> ----------- W2: proxy of THR ---------
>
> **Answer:** Thank you for this question. If we understand correctly, you are referring to Figure 3, where the correlation is between the magnitude (absolute value) of THR and entropy. However, this correlation does not imply that entropy can replace THR.
>
> **THR carries a sign**, while entropy is always positive and measures only local token confidence. In contrast, THR captures cross-token influence and indicates whether a token increases or decreases the likelihood of a correct response. This directional information is precisely what enables THR to control exploitation (p>0) versus exploration (p<0). Entropy alone cannot provide such behavioral control.
>
> Thus, although THR and entropy correlate in magnitude, the directional and relational information provided uniquely by the THR sign is essential for the controlled exploration–exploitation behavior studied in our work.

---

### Official Review · Reviewer_H57a · 2025-11-02

**Soundness:** 3
**Presentation:** 3
**Contribution:** 2
**Rating:** 4
**Confidence:** 3

**Summary:**

The paper proposes THR, a token-level signal derived from the training dynamics of GRPO-style RLVR that estimates whether a token will increase or decrease the likelihood of correct responses. The method operationalizes THR in two ways: (i) dominant-token selection that keeps only tokens with large |THR| and (ii) sign-based reweighting that tilts updates toward positive-THR tokens (exploitation) or negative-THR tokens (exploration). The stated goal is to provide a principled, fine-grained knob to trade off exploitation vs. exploration.

**Strengths:**

- The paper is anchored in a theoretical motivation (training-dynamics view under GRPO) rather than ad-hoc heuristics. It is intuitive that upweighting positive-THR tokens should push the model toward already promising trajectories, while emphasizing negative-THR tokens should diversify candidates.
- The structure of the paper is clean and logical, which makes it straightforward to map claims to evidence. Notation is compact, figures/tables are readable, and the roles of the sign-weighting scalar are explained sufficiently to re-implement. The exposition makes the method feel accessible to practitioners who already run GRPO/GSPO.

**Weaknesses:**

- The paper mentions training for ~40 steps, which is typically short for convergence in RLVR settings. From an evaluator’s perspective, this makes it hard to judge stability and whether gains persist or are transient. It would strengthen the paper to include training curves.
- Prior work (e.g., DAPO) shows that increasing clipping threshold (clip higher) can implicitly encourage exploration by preserving higher-entropy tokens. Given the paper’s token-level framing, a “clip-higher” baseline in Table 2 would be an informative.

**Questions:**

see weaknesses

---

> ### Author Response · Authors · 2025-11-21
> **Thank you very much!**
>
> Thank you for your thorough evaluation and helpful suggestions. We truly appreciate your time and address your comments point by point below.
>
> ---------- W1:  training steps -----------
>
> **Answer:** Thank you for your question. We would like to clarify a misunderstanding regarding the definition of steps.
>
> - **Effective Gradient Step:** We note that a “step” in our setup does not correspond to a single parameter update. We follow standard GRPO practice and with a prompt batch size of 256 and 8 rollouts per prompt. Then we use a mini-batch size of 64, resulting in 32 optimization iterations per step. Therefore, 40 steps corresponds to 40*32  = **1280 gradient steps**.
>
> - **Dynamic Sampling:** We would like to clarify that our training pipeline uses dynamic sampling, which reduces the steps by 3-4x [1] as all 0 and all 1 groups (which provide zero learning signal in the GRPO token-level objective) are discarded, and continually resample until a full effective batch is formed. This mechanism produces substantially more effective optimization steps than naïvely counting “training steps,” since it removes batches that would otherwise yield no gradient updates or small updates (small effective learning rate due to token-level loss normalization).  As a result, a more meaningful comparison is based on data consumed, not raw step count. Under this lens, the Qwen2.5-Math-1.5B model sees roughly two effective epochs (2.4 epochs), whereas the Qwen2.5-Math-7B model sees closer to three effective epochs of training.
>
> - **Convergence Curve:** To further illustrate convergence behavior, we report the accuracy on the MATH (levels 3–5) validation set throughout training for Qwen2.5-Math-1.5B and Qwen2.5-Math-7B in Appendix Fig. 11 of the revised version. The validation accuracy increases steadily until approximately 30–35 steps, after which the gains plateau, indicating that the models have largely converged. Although additional training steps might provide small further improvements, they also increase the risk of overfitting. Since our objective is to highlight the trend and behavioral effects of p rather than maximize absolute accuracy, we standardize the training budget to 40 steps across all settings to ensure consistency and fair comparison.
>
> [1] DAPO: An open-source LLM reinforcement learning system at scale.
>
> --------- W2: Compared with CLIP higher---------
>
> **Answer:** Thank you for your question. Following your suggestion, we added a clip-higher baseline using the same threshold (0.28) in [1] on Qwen2.5-Math-1.5B. As shown in the table below, clip-higher indeed improves exploration for K ≥ 32 over GRPO.
>
> | **Method** | **1** | **2** | **4** | **8** | **16** | **32** | **64** | **128** | **256** |
> |------------|-------|-------|-------|-------|--------|--------|--------|---------|---------|
> | **AIME 2025** |||||||||||
> | GRPO | 5.9 | 9.9 | 15.0 | 20.5 | 26.5 | 33.6 | 41.5 | 49.8 | 56.7 |
> | Clip-High | 5.6 | 9.6 | 14.8 | 20.5 | 26.6 | 33.7 | 41.6 | 48.4 | 53.3 |
> | THR ($p<0$) | **6.0** | **10.1** | **15.3** | **20.9** | **26.8** | **33.9** | **41.7** | **50.0** | **60.0** |
> | **AIME 2024** |||||||||||
> | GRPO | 11.4 | 17.7 | 24.3 | 30.5 | 36.7 | 43.4 | 50.0 | 56.0 | 63.3 |
> | Clip-High | 10.8 | 16.7 | 23.2 | 29.8 | 36.5 | 44.0 | 52.1 | 60.7 | 70.0 |
> | THR ($p<0$) | **11.9** | **18.2** | **24.9** | **31.2** | **37.9** | **45.3** | **52.9** | **61.2** | **70.0** |
> | **AMC 2023** |||||||||||
> | GRPO | 46.6 | 59.1 | 70.0 | 78.9 | 85.5 | 90.2 | 93.7 | 96.0 | 97.5 |
> | Clip-High | 47.3 | 59.9 | 70.5 | 78.8 | 84.9 | 89.8 | 93.8 | 97.3 | 100.0 |
> | THR ($p<0$) | **47.9** | **61.0** | **72.2** | **81.1** | **87.3** | **91.6** | **95.1** | **98.0** | **100.0** |
> | **Average** |||||||||||
> | GRPO | 21.3 | 28.9 | 36.4 | 43.3 | 49.6 | 55.7 | 61.7 | 67.3 | 72.5 |
> | Clip-High | 21.2 | 28.7 | 36.2 | 43.0 | 49.3 | $\underline{55.8}$ | $\underline{62.5}$ | $\underline{68.8}$ | $\underline{74.4}$ |
> | THR ($p<0$) | **21.9** | **29.8** | **37.5** | **44.4** | **50.7** | **57.3** | **63.2** | **69.7** | **76.7** |
>
> Despite being a strong baseline, THR (p<0) still consistently outperforms clip-higher across all K, with gains often exceeding 1%. This consistent improvement shows the effectiveness of THR (p<0) for exploration.
>
> We appreciate the reviewer’s suggestion and have included this baseline and discussion in the revised version.

---

### Official Review · Reviewer_KUZ2 · 2025-11-03

**Soundness:** 3
**Presentation:** 3
**Contribution:** 3
**Rating:** 6
**Confidence:** 3

**Summary:**

The paper introduces Token Hidden Reward (THR), a token-level metric that quantifies each token’s impact on correct-response likelihood in popular GRPO frameworks. The authors show that a small subset of high-THR tokens dominates training, enabling fine-grained control of the exploration–exploitation trade-off: positive THR drives exploitation, while negative THR promotes exploration. The proposed THR-guided reweighting steers learning toward exploration/exploitation, improving greedy accuracy (p > 0)/Pass@K (p < 0) on math reasoning benchmarks.

**Strengths:**

- The introduced THR framework is principled and fundamental, and connects learning dynamics in GRPO-tuned LLMs to the exploration–exploitation trade-off. This formulation offers a conceptually grounded and theoretically principled perspective that deepens understanding of how token-level interactions shape model behavior during reinforcement learning.

- The paper is generally well written, logically structured, and clearly motivated, making it easy to follow. The authors conduct extensive experimental studies across multiple model scales (0.5B–7B) and architectures (Qwen and Llama), complemented by additional analyses and ablation experiments. Collectively, these results provide solid empirical support for the proposed approach and its underlying claims (while there are some inconsistencies and potential concerns; see weakness section).

**Weaknesses:**

- The empirical improvements reported between variants and baselines are relatively modest. Incorporating statistical significance analyses would help validate the robustness of these gains and ensure that the observed effects are not attributable to stochastic randomness.

- The reported results indicate that the optimal choice of the hyperparameter p varies across model sizes and model families (e.g., Qwen vs. Llama), suggesting that the method’s efficacy may depend strongly on careful tuning. A more systematic discussion or ablation study on how p is selected—and whether its value generalizes across architectures or tasks—would strengthen the methodological soundness (please correct me if I missed such discussion).

- In Table 1, the observation that p > 0 fails to consistently outperform p < 0 for smaller models (0.5B and 1.5B) appears inconsistent with the claimed interpretation that positive p promotes exploitation. This discrepancy weakens the theoretical linkage between the sign of p and behavioral control.

- While experiments on Llama are included to demonstrate the generalizability of the proposed THR method, the reported results appear to hold primarily for the exploration configuration (p < 0). The exploitation experiments on Llama (Table 8) show that THR with p > 0 underperforms the GRPO baseline, suggesting that the proposed mechanism may not generalize robustly across architectures.

**Questions:**

- Why does training on THR-dominant tokens yield comparable, instead of better performance compared to GRPO given its potential effectiveness in proving stronger learning signals to guide the training process? Alternatively, does it lead to faster convergence?

---

> ### Author Response · Authors · 2025-11-21
> **Thank you! (part 1)**
>
> Thank you for your thorough evaluation and helpful suggestions. We truly appreciate your time and address your comments point by point below.
>
> --------- W1: empirical improvements ----------
>
> **Answer:**  We respectfully disagree that the improvements are modest. The observed gains are both substantial in magnitude and consistent across model scales and architectures, confirming that the effects are not due to stochastic randomness.
>
> *Magnitude of improvements.* For exploitation, Qwen2.5-Math-7B with p>0 improves greedy accuracy by +4.1% over GRPO and +3.5% over THR(p=0). On smaller models (0.5B and 1.5B), the gains remain consistent, exceeding +1.5%. For exploration, the improvement is also pronounced: on Qwen2.5-Math-1.5B, Pass@K increases by up to +4%, and on Llama3.2-3B, THR(p<0) achieves roughly +10% improvement over GRPO at K=256. These are not minor fluctuations but clear, directional performance gains.
>
> *Directional consistency across all K and scales indicates a robust effect, not noise.*  THR(p<0) consistently surpasses both GRPO and THR(p=0) across all K values, typically by more than 1 % in all K. This monotonic trend across Qwen 0.5B–7B and Llama 3B demonstrates that the observed effects are systematic and scale-consistent, rather than random.
>
> *Deterministic and low-variance evaluation.* Greedy decoding accuracy is deterministic, and Pass@K scores are computed from 256 independent rollouts, ensuring stable and low-variance estimates.
>
> ------- W2: systematic ablation study on p ------
>
> **Answer:**   Thank you for your question. We would like to clarify that we did not perform extensive hyperparameter tuning, as the primary goal of this work is to demonstrate that a simple theory-driven adjustment of the parameter p can reliably steer the model toward exploration or exploitation.
>
> As noted in the manuscript, we use the same default values across most settings, specifically, p=0.1 for exploitation and p=−0.1 for exploration. The only exception is Qwen2.5-0.5B-Instruct, where we observed that p=−0.2 yielded stronger improvements, and we chose p=0.2 for symmetry in the exploitation counterpart. These fixed settings already yield consistent and significant improvements for Qwen2.5-Math-1.5B, Qwen2.5-Math-7B, and Llama3.2-3B-Ins, as shown in Table 1, Table 2, and Table 9.
>
> That said, following your suggestion, we conducted a sensitivity analysis on p using **Qwen2.5-Math-1.5B**.
>
> (1) *For exploration*, we use p = [0,-0.05,-0.1,-0.2]. We observe a consistent exploration trend as shown in the table below (complete results in Table 12 of the revised appendix ), where setting p<0 can consistently improve the pass@K performance, thus reinforcing the conclusion that p<0 reliably enhances exploration.
>
> | Method | 1 | 2 | 4 | 8 | 16 | 32 | 64 | 128 | 256 |
> |-------|---|---|---|---|----|----|----|-----|-----|
> | **Average** |||||||||||
> | GRPO | 21.3 | 28.9 | 36.4 | 43.3 | 49.6 | 55.7 | 61.7 | 67.3 | 72.5 |
> | THR | 20.3 | 28.0 | 35.5 | 42.6 | 49.1 | 55.5 | 61.7 | 67.5 | 72.5 |
> | THR (p = −0.05) | $\underline{22.0}$ | 29.6 | 37.2 | 43.9 | 50.1 | 56.4 | 62.8 | 69.4 | 76.7 |
> | THR (p = −0.1) | 21.9 | $\underline{29.8}$ | $\underline{37.5}$ | **44.4** | $\underline{50.7}$ | $\underline{57.3}$ | $\underline{63.2}$ | $\underline{69.7}$ | **76.7** |
> | THR (p = −0.2) | **22.8** | **30.3** | **37.6** | $\underline{44.3}$ | **51.0** | **58.3** | **65.3** | **70.7** | 74.4 |
>
> We also examined the effect of p on Llama. As shown in Fig. 5 and Table 9, setting p=−0.1  or −0.2 consistently improves exploration, with p=−0.2 achieving a +10% gain over GRPO at larger K=256. This result highlights that, even though Llama3.2-3B is a non-reasoning model with shorter response lengths (Fig. 6), applying p<0 still reliably enhances exploration.

---

> ### Author Response · Authors · 2025-11-21
> **Thank you very much (Part 2)**
>
> (2) *For exploitation*, we evaluated p∈[0,0.05,0.1,0.2]. The results in the table below show that reducing p from 0.1 to 0.05 yields the best greedy accuracy, outperforming GRPO by 2.8% and p<0 by 1.2%, indicating that a milder exploitation strength can be more effective for the weaker qwen2.5-Math-1.5B model. Conversely, increasing p to 0.2 slightly decreases greedy accuracy relative to p=0.1,  likely due to over-emphasizing exploitation; however, its performance remains above the alternative settings, confirming that the exploitation effect is still preserved.
>
> | **Method** | AIME25 | AIME24 | AMC23 | Hard Avg. | MATH500 | Minerva | Olympiad | Standard Avg. | Total Avg. |
> |------------|--------|--------|--------|-----------|---------|---------|----------|----------------|------------|
> | Base | 0.0 | 3.3 | 20.0 | 7.8 | 39.6 | 7.7 | 24.9 | 24.1 | 15.9 |
> | GRPO | 3.3 | **13.3** | 57.5 | 24.7 | **71.8** | 29.0 | $\underline{34.1}$ | 45.0 | 34.8 |
> | THR | 3.3 | **13.3** | 55.0 | 23.9 | 70.8 | 32.4 | $\underline{34.1}$ | 45.8 | 34.8 |
> | THR ($p=-0.1$) | **10.0** | **13.3** | 60.0 | $\underline{27.8}$ | 70.6 | 32.0 | 32.7 | 45.1 | $\underline{36.4}$ |
> | THR ($p=0.05$) | **10.0** | **13.3** | **62.5** | **28.6** | **71.8** | **35.7** | 32.1 | **46.5** | **37.6** |
> | THR ($p=0.1$) | 3.3 | **13.3** | **62.5** | 26.4 | 71.4 | $\underline{33.1}$ | **34.5** | $\underline{46.3}$ | 36.3 |
> | THR ($p=0.2$) | 3.3 | **13.3** | 60.0 | 25.5 | 71.0 | 32.7 | 33.9 | 45.9 | 35.7 |
>
>
> --------- W3: Exploitation on Smaller Models ------
>
> **Answer:**  Thank you for your question. We respectfully disagree that the results for smaller models indicate an inconsistency. The preference for p<0 arises not because the theory breaks, but because smaller models have limited capability, thus exploration could be beneficial. In contrast, p>0 consistently enhances exploitation once the model is strong enough or when a milder exploitation strength is used.
> - Standard benchmarks confirm the expected exploitation effect.  On the standard datasets, both Qwen2.5-0.5B and Qwen2.5-1.5B exhibit higher greedy accuracy with p>0 than with p<0 (Table 1). This is exactly the expected effect of p>0: reinforcing correct observed responses to strengthen exploitation.
> - Hard-dataset behavior reflects limited model capability, not inconsistency. For smaller models, p<0 only performs better on hard datasets where their base accuracy is extremely low (0.8% for 0.5B and 7.8% for 1.5B). In such low-knowledge regimes, exploration (p<0) can be helpful, while a strong exploitation (p=0.1) can be ineffective without reliable solution modes to exploit. In contrast, when the model is stronger, e.g., Qwen2.5-Math-7B with a hard average of 20.8%, p>0 yields consistent and substantial gains (e.g., +3.3% over p<0). This is fully aligned with our theory: p>0 enhances exploitation, whereas p<0 enhances exploration.
> - Using a smaller exploitation strength. By reducing p to p=0.05 on Qwen2.5-Math-1.5B, we achieve better greedy accuracy, outperforming GRPO by +2.8% and p<0 by +1.2%. This shows that p=0.1 was simply too strong for a small model; weaker models still benefit from exploitation, but with a milder positive p.
>
> Once more, we wish to clarify that we didn’t conduct careful hyperparameter tuning, as the goal of this work is to show that a simple adjustment of p can reliably steer the model toward exploration or exploitation. The expected behaviors emerge consistently even without tuning.
>
> --------- W4: Llama Model ---------
>
> **Answer:**  Thank you for your question. As noted in Section 6.4, Llama3.2-3B is a non-reasoning model that exhibits reduced reasoning length during training (Fig. 6).  Despite this, we still observe a consistent directional effect of p on Llama. Specifically, p>0 improves greedy exploitation performance compared to p=0 by  +1.1 % and to  p<0 by + 3.4%, demonstrating that the exploitation conclusion is consistent. More importantly, Llama shows strong and stable exploration gains under p<0. THR(p=-0.2) achieves up to +10 % improvement over GRPO at K=256, highlighting that THR generalizes well to Llama.
>
> Thus, we respectfully disagree that the proposed mechanism fails to generalize across architectures. Instead, the Llama results support the directional control of p: p>0 for exploitation and p<0 for exploration, remains consistent across all model families evaluated.

---

> ### Author Response · Authors · 2025-11-21
> **Thank you very much (Part 3)**
>
> --------- Q1: THR dominant didn’t outperform GRPO ----------
>
> **Answer:** Thank you for the question. Our goal with the THR-dominant token experiment is **not** to outperform GRPO, but to demonstrate that **a very small subset of high-influence tokens almost determines the learning signal in GRPO**. As noted in lines 313-316 and Table 1, training only on THR-dominant tokens yields performance comparable to full GRPO, despite discarding most of the tokens.  This result provides two meaningful insights:
>
> - Effectiveness of THR: Matching GRPO with only dominant tokens confirms that THR accurately identifies the tokens that matter most for driving training dynamics. If training on a small amount of tokens recovers the same performance, this strongly validates the importance of THR as an influence metric.
>
> - Computational and memory efficiency: Because THR-dominant training ignores 70–80% of tokens, it opens the door to efficiency gains. By stopping gradient computation on non-dominant tokens, memory consumption can be reduced, and training throughput can be decreased. This is a practical advantage rather than a limitation.

---

### Meta-Review · Area_Chair_hqkD · 2025-12-23

**Summary:**

This paper introduces Token Hidden Reward (THR), a token-level reward signal designed to steer the exploration–exploitation balance in GRPO for language models. By assigning positive or negative reward signals to tokens based on whether they increase or decrease the likelihood of correct responses, the THR method provides a fine-grained and interpretable mechanism for guiding RL training dynamics. Reviewers broadly agree that the idea is well motivated, theoretically grounded, and clearly connected to practical RL training pipelines for LLMs.

Across review comments, the paper is considered as making a meaningful conceptual contribution by moving beyond sequence-level rewards. The empirical evaluation is extensive, including multiple models, benchmarks, ablation studies, and shows improved exploration control and accuracy compared to GRPO and other baselines. The rebuttal further strengthened the work by adding missing baselines, clarifying computational overhead, and addressing concerns about robustness and generalization.

**Reviewer Concerns:**

### Concerns addressed by the rebuttal: ###

- The authors clarified that improvements are consistent across model scales and datasets, added variance-aware analyses, and showed that gains are not due to stochastic noise.

- Additional ablations and comparisons (e.g., clip-higher entropy baselines, scaling across different K values) helped demonstrate that THR specifically improves exploration rather than simply increasing randomness.

- The rebuttal provided results showing that THR introduces minimal additional cost compared to standard GRPO training.

- Several reviewers requested stronger or more direct baselines, where these were added and discussed during rebuttal.

-The revised paper version improved explanations of THR computation, its relation to entropy, and practical integration into existing RL frameworks.

### Remaining concerns: ###

- Some reviewers noted that the method is primarily evaluated in math and reasoning-heavy domains, and broader validation on non-reasoning tasks could further strengthen the paper.

- A few points about presentation clarity may remain, but these are minor and do not affect the core contribution.

**Reviewer Scores:**

- Reviewer KUZ2 (Original: 6): The rebuttal directly addressed concerns about statistical robustness, scaling behavior, and baseline completeness. With the added experiments and clarifications, this reviewer would likely increase rating.

- Reviewer H57a (Original: 4): The main concerns were about interpretability, stability of RL dynamics, and whether token-level rewards generally improve exploration rather than acting as an entropy proxy. The rebuttal directly addressed these points with additional baselines (e.g., clip-higher entropy) and clearer explanation of THR’s directional signal. With these clarifications, this reviewer may increase rating.

- Reviewer zvLX (Original: 6): Questions about computational cost and the relationship between THR and entropy were convincingly addressed, which likely increases confidence in the method’s practicality.

- Reviewer on7i (Original: 6): The added ablations, stronger baselines, and clearer explanation of why token-level rewards matter for exploration-exploitation balance would likely raise this reviewer’s rating.

---

### Decision · Program_Chairs · 2026-01-26

Accept (Poster)